# A microtubule-LUZP1 association around tight junction promotes epithelial cell apical constriction

Tomoki Yano[1,2,†] [ID], Kazuto Tsukita[2,3,†] [ID], Hatsuho Kanoh[2,4,†], Shogo Nakayama[2], Hiroka Kashihara[2] [ID], Tomoaki Mizuno[2], Hiroo Tanaka[2,5,6], Takeshi Matsui[7] [ID], Yuhei Goto[8,9,10], Akira Komatsubara[8,9,10], Kazuhiro Aoki[8,9,10] [ID], Ryosuke Takahashi[3], Atsushi Tamura[2,5,6,*] [ID] & Sachiko Tsukita[2,6,**] [ID]

## Abstract

Apical constriction is critical for epithelial morphogenesis, including neural tube formation. Vertebrate apical constriction is induced by di-phosphorylated myosin light chain (ppMLC)-driven contraction of actomyosin-based circumferential rings (CRs), also known as perijunctional actomyosin rings, around apical junctional complexes (AJCs), mainly consisting of tight junctions (TJs) and adherens junctions (AJs). Here, we revealed a ppMLC-triggered system at TJ-associated CRs for vertebrate apical constriction involving microtubules, LUZP1, and myosin phosphatase. We first identified LUZP1 via unbiased screening of microtubule-associated proteins in the AJC-enriched fraction. In cultured epithelial cells, LUZP1 was found localized at TJ-, but not at AJ-, associated CRs, and LUZP1 knockout resulted in apical constriction defects with a significant reduction in ppMLC levels within CRs. A series of assays revealed that ppMLC promotes the recruitment of LUZP1 to TJ-associated CRs, where LUZP1 spatiotemporally inhibits myosin phosphatase in a microtubule-facilitated manner. Our results uncovered a hitherto unknown microtubule-LUZP1 association at TJ-associated CRs that inhibits myosin phosphatase, contributing significantly to the understanding of vertebrate apical constriction.

**Keywords** actomyosin-based circumferential rings; apical constriction; apical microtubules; LUZP1; tight junction
**Subject Categories** Cell Adhesion, Polarity & Cytoskeleton; Post-translational Modifications & Proteolysis
**The EMBO Journal (2021) 40: e104712**

## Introduction

Epithelial cells adhere to each other to form epithelial cell sheets. Apical constriction is a process wherein the apical side of an individual epithelial cell constricts to alter its morphology from columnar to wedge-shaped; therefore, apically constricted individual epithelial cells collectively induce epithelial cell sheet folding, which is crucial for many biological processes including gastrulation and neural tube formation (Sawyer *et al*, 2010; Suzuki *et al*, 2012; Martin & Goldstein, 2014; Takeichi, 2014; Krueger *et al*, 2018). Indeed, knockout (KO) of molecules involved in apical constriction often results in embryonic lethality and/or neural tube closure defects (NTDs) (Gates *et al*, 2007; Copp & Greene, 2010; David *et al*, 2010; Nikolopoulou *et al*, 2017), highlighting the importance of understanding the underlying regulatory mechanism of apical constriction.

Myosin activation within actomyosin filaments that are linked to cell membranes via apical cell–cell junctions provides the contracting force required to drive apical constriction of individual epithelial cells (Martin & Goldstein, 2014; Hunter & Fernandez-Gonzalez, 2017). It should be noted that the different patterns of apical actomyosin arrangements are dominantly assembled for apical constriction in invertebrates or vertebrates. In invertebrates like *Drosophila* and *Caenorhabditis elegans*, the medioapical actomyosin accumulation in the middle of the apical area is actively involved in apical constriction (Martin & Goldstein, 2014). On the other hand, actomyosin-based circumferential rings (CRs) associated with apical junctional complexes (AJCs), which includes tight junctions (TJs) and adherens junctions (AJs), play a central role in vertebrate apical constriction (Sawyer *et al*, 2010; Martin & Goldstein, 2014; Takeichi, 2014), although there are indications that the medioapical actomyosin network still plays a role in some occasions (Sumigray *et al*,

1 Laboratory of Biological Science, Graduate School of Medicine, Osaka University, Osaka, Japan
2 Laboratory of Barriology and Cell Biology, Graduate School of Frontier Biosciences, Osaka University, Osaka, Japan
3 Department of Neurology, Graduate School of Medicine, Kyoto University, Kyoto, Japan
4 Graduate School of Biostudies, Kyoto University, Kyoto, Japan
5 Department of Pharmacology, School of Medicine, Teikyo University, Tokyo, Japan
6 Strategic Innovation and Research Center, Teikyo University, Tokyo, Japan
7 Laboratory for Skin Homeostasis, Research Center for Allergy and Immunology, RIKEN Center for Integrative Medical Sciences, Kanagawa, Japan
8 Exploratory Research Center on Life and Living Systems (ExCELLS), National Institutes of Natural Sciences, Aichi, Japan
9 National Institute for Basic Biology, National Institutes of Natural Sciences, Aichi, Japan
10 Department of Basic Biology, Faculty of Life Science, SOKENDAI (Graduate University for Advanced Studies), Aichi, Japan
*Corresponding author. Tel: +81 6 6879 4633; E-mail: atamuta@biosci.med.osaka-u.ac.jp
**Corresponding author. Tel: +81 6 6879 4633; E-mail: atsukita@biosci.med.osaka-u.ac.jp, SachikoTsukita3@gmail.com
†These authors contributed equally to this work

2018). Myosin activation primarily results from phosphorylation of myosin light chain (MLC) and, while MLC is either mono- or di-phosphorylated at T18 and/or S19, di-phosphorylated MLC (ppMLC) is particularly critical for actomyosin contraction (Miyake et al, 2006; Watanabe et al, 2007). Therefore, MLC phosphorylation status, especially ppMLC levels, within AJC-associated CRs determines the contracting force within CRs and is the critical regulatory step for vertebrate apical constriction. Accordingly, the mechanism underlying modulation of the localization and/or function of AJC-localized Rho-associated coiled-coil kinase (ROCK), the primary driving kinase promoting MLC phosphorylation, has been extensively studied. For example, it was shown that shroom3, whose AJC localization is modulated by other factors such as Trio and Lulu (Nakajima & Tanoue, 2011; Plageman et al, 2011; Chu et al, 2013), recruits ROCK to AJCs (Hildebrand & Soriano, 1999; Nishimura & Takeichi, 2008), whereas the willin/Par3-atypical protein kinase C (aPKC) pathway suppresses AJC localization of ROCK (Ishiuchi & Takeichi, 2011). It has been also shown that planar cell polarity modulates ROCK function by upregulating PDZ-Rho GEF (Nishimura et al, 2012). However, given that more than 300 genes have been reported to cause NTDs, the regulatory mechanism of apical constriction appears highly sophisticated and far from complete elucidation (Copp & Greene, 2010; Nikolopoulou et al, 2017; Krueger et al, 2020; Denk-Lobnig & Martin, 2020).

The balance between kinases and phosphatases acting on MLC determines its phosphorylation status. Myosin phosphatase, which is a hetero-trimer consisting of protein phosphatase 1c $\beta/\delta$ (PP1c $\beta/\delta$), myosin phosphatase targeting subunit 1 (MYPT1), and a small 20-kDa regulatory subunit (M20) (Kiss et al, 2019), critically regulates MLC phosphorylation status by downregulating its phosphorylation. However, in the context of apical constriction, limited information regarding the regulatory mechanism of myosin phosphatase is available. ROCK also partially contributes to the inhibition of myosin phosphatase by phosphorylating MYPT1 in addition to directly phosphorylating MLC (Jain et al, 2018; Kiss et al, 2019); however, it remains unknown whether a mechanism that primarily regulates myosin phosphatase activity exists. Another unanswered question of vertebrate apical constriction is how microtubules (MTs) are involved. Previous studies on the vertebrate Xenopus laevis have revealed that MT polymerization inhibitors as well as the loss of MT-stabilizing factors, namely MID1 and MID2, lead to apical constriction defects (Lee & Harland, 2007; Suzuki et al, 2010). However, the mechanism underlying the indispensable role of MTs in vertebrate apical constriction remains unknown.

Here, we provide conceptual advances by revealing a crucial ppMLC-triggered system at CRs around TJs—hereafter referred to as TJ-associated CRs (Fig EV1A)—for vertebrate apical constriction involving LUZP1, myosin phosphatase, and MTs. The apical MTs, which are different from classical apicobasal MTs, are reportedly associated with TJs in a side-by-side manner, in a variety of epithelial cells (Fig EV1B) (Kunimoto et al, 2012; Yano et al, 2013; Matter & Balda, 2014; Herawati et al, 2016; Toya & Takeichi, 2016; Yano et al, 2017; Takeda et al, 2018; Yano et al, 2018; Citi, 2019; Tsukita et al, 2019a). To further elucidate the role of MTs in cell–cell junctions, we conducted unbiased screening for MT-associated proteins in an AJC-enriched fraction and identified LUZP1. LUZP1 was previously proposed to play an important role in apical constriction because its KO leads to cranial NTDs in mice through an

unidentified mechanism (Hsu et al, 2008). Using super-resolution immunofluorescence microscopy and immunoelectron microscopy, we found that LUZP1 was predominantly localized at TJ-associated CRs, not AJ-associated CRs, in cultured epithelial Eph4 cells. Co-cultures of LUZP1 KO and LUZP1-expressing cells (wild-type [WT] cells or Venus-LUZP1-expressing LUZP1 KO [REV] cells) revealed that LUZP1 KO leads to apical constriction defects with significantly downregulated ppMLC levels within CRs. A series of assays revealed that the recruitment of LUZP1 to TJ-associated CRs is promoted by ppMLC owing to strong binding activity between LUZP1 and ppMLC (dissociation constant $[K_d] < 1$ $\mu$M), where LUZP1 inhibits myosin phosphatase by suppressing the activity of its catalytic subunit, PP1c $\beta/\delta$, in an MT-facilitated manner. Altogether, our findings revealed that the ppMLC-triggered, MT-facilitated, and LUZP1-based system spatiotemporally inhibits myosin phosphatase at TJ-associated CRs and is thereby critical for maintaining ppMLC levels within CRs to promote vertebrate apical constriction.

## Results

### LUZP1 is an MT-associated protein that localizes at TJ-associated CRs

We previously identified four MT-binding proteins in the AJC fraction prepared from chick livers via membrane overlay assays of Taxol-stabilized MTs (Tsukita & Tsukita, 1989; Yamazaki et al, 2008; Yano et al, 2013). Here, we identified one of these proteins as LUZP1 from its amino acid sequence (Figs 1A and EV1C). To confirm the interaction between LUZP1 and MTs, we performed MT co-sedimentation assays and found that Flag-LUZP1 directly bound MTs (dissociation constant $[K_d] = 0.78 \pm 0.13$ $\mu$M; Fig 1B). Further MT co-sedimentation assays revealed that N-terminal and middle regions were responsible for binding to MTs (Fig EV1D and E). Next, we generated antibodies against the N-terminal, middle, and C-terminal regions of LUZP1. The specificities of these antibodies were confirmed by the lack of their immunoblot signals in LUZP1 KO mouse mammary gland epithelial Eph4 cells, which were also generated (Fig EV1F). Immunoblotting using these antibodies confirmed that LUZP1 was enriched in the AJC fraction (Fig EV1G) and showed that LUZP1 was ubiquitously expressed in various tissues (Fig EV1H).

Immunofluorescent staining of the cultured epithelial Eph4 cells revealed that LUZP1 was preferentially associated with cell–cell junctions at the level of TJs, which are positive for ZO-1 (Fig 1C and Movie EV1). Super-resolution micrographs further revealed that LUZP1 was localized as two separate parallel lines along the single ZO-1-positive lines, suggesting that LUZP1 is distributed around TJs where actin and myosin form CRs (Fig 1D and E). A close association between apical MTs and LUZP1 was also observed, altogether indicating that LUZP1 was localized at TJ-associated CRs where MTs are also enriched (Fig 1F). Co-immunoprecipitation assays of WT Eph4 cells using the anti-LUZP1 antibody indicated that LUZP1 simultaneously associates with ZO-1 (a TJ protein), MLC (a CR constituent), and $\alpha$-tubulin (Fig EV1I), supporting immunofluorescence observations. To further confirm the localization of LUZP1, we generated REV cells by transfecting Venus-LUZP1 into LUZP1 KO cells and confirmed that exogenous Venus-LUZP1 was similarly localized (Fig EV2A). We also performed immunoelectron

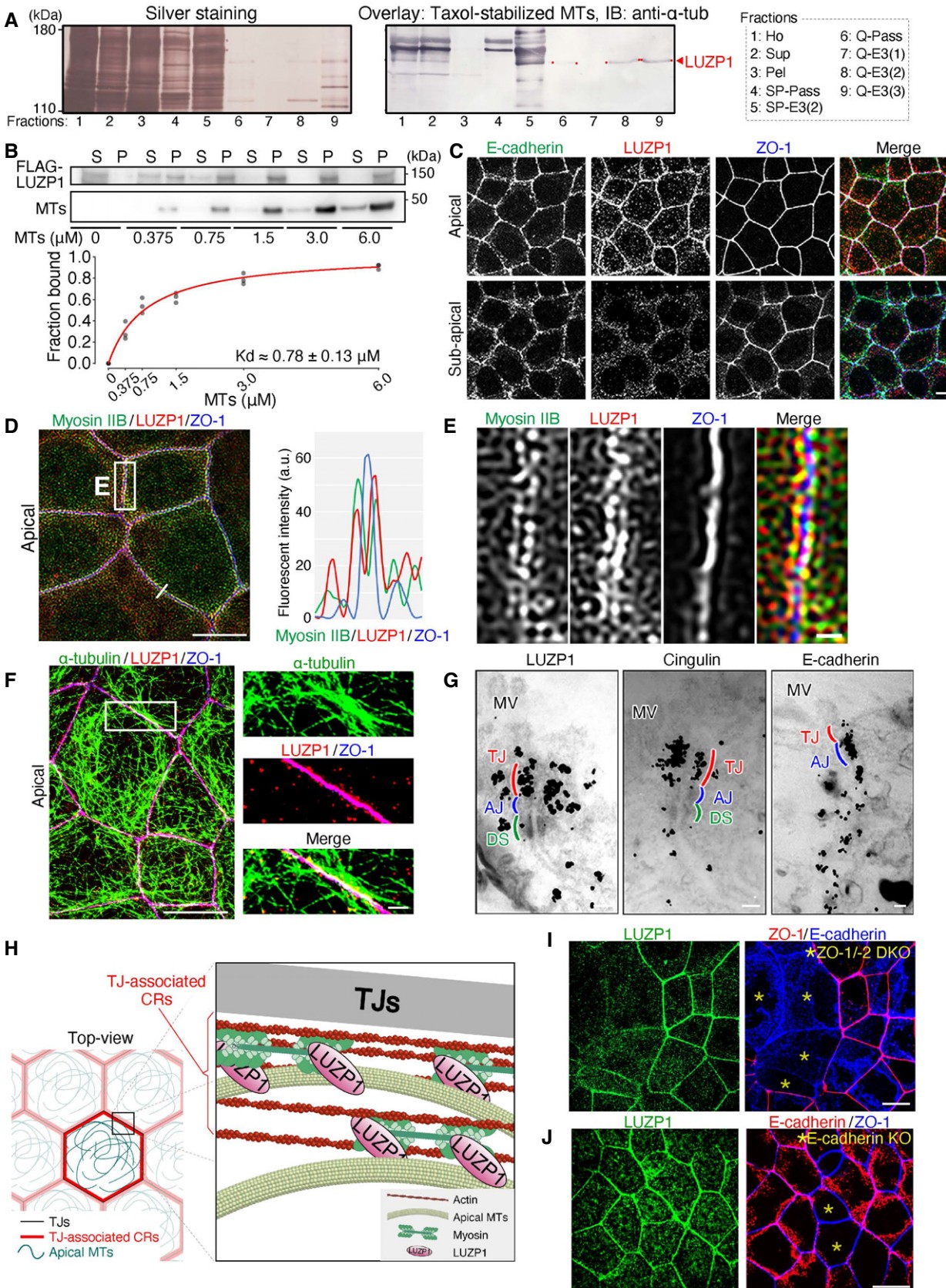

**Figure 1.**

**Figure 1.  LUZP1 is a microtubule (MT)-binding protein localized at tight junction (TJ)-associated circumferential rings (CRs).**

A  Membrane overlay assay of Taxol-stabilized MTs using the apical junctional complex (AJC)-enriched fraction. The AJC-enriched fraction was further fractionated using ultra-centrifugation and ion-exchange chromatography with an SP-Sepharose and a Q-Sepharose column (see also Fig EV1C). The bands (marked by an arrowhead and red dots) correspond to LUZP1. 1: Ho, homogenate; 2: Sup, supernatant; 3: Pel, pellet; 4: SP-Pass, passing material through an SP-Sepharose column; 5: SP-E3(2), $2^{nd}$ elution fraction through an SP-Sepharose column; 6: Q-Pass, passing material through a Q-Sepharose column; 7–9: Q-E3(1) to Q-E3(3), $1^{st}$ to $3^{rd}$ elution fractions through a Q-Sepharose column.

B  MT co-sedimentation assay with purified Flag-LUZP1. Flag-LUZP1 directly bound MTs with a dissociation constant ($K_d$) of 0.78 ± 0.13 μM. A representative immunoblot and a plot in which all results are plotted with the Michaelis–Menten fitted curve are shown. $n = 3$. S, supernatant; P, pellet.

C  Representative confocal micrographs of immunostained Eph4 epithelial cell sheets in the apical and sub-apical planes. LUZP1 was associated with TJs which are positive for ZO-1. Scale bar, 10 μm.

D  A representative super-resolution micrograph of immunostained Eph4 epithelial cell sheets with the graph showing fluorescent intensities along the line. LUZP1 co-localized better with myosin IIB than with ZO-1. a.u., arbitrary units. Scale bar, 10 μm.

E  Magnified images of Fig 1D. Scale bar, 1 μm.

F  Representative confocal micrographs of immunostained Eph4 epithelial cell sheets in the apical plane. The apical MTs were associated with both LUZP1 and TJs. Scale bar, 10 μm (low magnification) and 1 μm (high magnification).

G  Representative immunoelectron micrographs of Eph4 cells. Immunogold particles for LUZP1 preferentially accumulated at the level of TJs that are positive for cingulin. MV, microvilli; AJ, adherens junction, DS, desmosome. Scale bars, 200 nm.

H  A schematic drawing of LUZP1 localization in association with TJs, apical MTs, and actomyosin-based CRs.

I  Representative confocal micrographs of co-cultures of wild-type (WT) and ZO-1/-2 double knockout (DKO) Eph4 cells. ZO-1/-2 DKO cells were marked by asterisks (*). The LUZP1 junctional localization was apparently disrupted in TJ-deficient ZO-1/-2 DKO cells. Scale bar, 10 μm.

J  Representative confocal micrographs of transient E-cadherin knockout (KO) in WT Eph4 cells. E-cadherin KO cells were marked by asterisks (*). The LUZP1 junctional localization seemed not to change between AJ-deficient E-cadherin KO and WT cells. Scale bar, 10 μm.

Source data are available online for this figure.

---

microscopic analyses to analyze LUZP1 localization more precisely. Consistent with our immunofluorescence observations, the accumulation pattern of immunogold particles for LUZP1 was detected in CR-regions, at the same level of those of TJ-related proteins (cingulin, occludin, and claudin-7), but different from those of AJ-related proteins (E-cadherin and β-catenin; Figs 1G and EV2B). Finally, we examined LUZP1 localization in mouse tissues, including the embryonic neural tube and small intestine, and confirmed that LUZP1 was also distributed as two separate lines around TJs in *in vivo* tissues (Fig EV2C and D). Collectively, these results show that LUZP1 is an MT-binding protein localizing at TJ-associated CRs in both cultured and *in vivo* epithelial cells (Fig 1H).

To assess the importance of TJs in LUZP1 junctional localization, we generated ZO-1/-2 double knockout (DKO) cells which are known to be TJ-deficient (Umeda *et al*, 2006; Ikenouchi *et al*, 2007; Otani *et al*, 2019). Subsequently, we found that, even in these TJ-deficient ZO-1/-2 DKO cells which have intact AJs, LUZP1 junctional localization was apparently impaired (Figs 1I and EV2E), confirming the crucial role of TJs in the recruitment of LUZP1 to TJ-associated CRs. We next transiently knocked out E-cadherin in WT cells to disrupt AJs and found that E-cadherin KO did not influence LUZP1 junctional localization (Fig 1J), further confirming that LUZP1 junctional localization is regulated by TJs, not AJs, in AJCs. Co-immunoprecipitation analyses revealed that LUZP1 bound both ZO-1 and ZO-2 (Fig EV2F and G), suggesting that binding affinity for these TJ-related proteins are responsible for recruiting LUZP1 to TJ-associated CRs.

**LUZP1 at the TJ-associated CRs is crucial for ppMLC upregulation within CRs to induce apical constriction**

To explore the function of LUZP1 in epithelial cells, we next examined the phenotypes of LUZP1 KO epithelial Eph4 cells. Co-cultures of LUZP1 KO and REV cells showed that the apical area became smaller than the basal area only in REV cells (Fig 2A and B). Co-

cultures of LUZP1 KO and WT cells revealed that the apical area became significantly smaller in WT cells than in LUZP1 KO cells (Fig 2C and D), altogether indicating that LUZP1 KO cells have apical constriction defects. We next examined the influence of LUZP1 KO on actomyosin organization and ppMLC levels within CRs because apical constriction is reported to be primarily regulated by the actomyosin organization of CRs and/or myosin activation within CRs in vertebrates (Sawyer *et al*, 2010; Suzuki *et al*, 2012; Martin & Goldstein, 2014; Takeichi, 2014). Immunofluorescence analyses showed that LUZP1 KO had no effect on actomyosin organization of CRs (Fig EV3A and B); however, ppMLC levels within CRs were clearly and significantly downregulated in LUZP1 KO Eph4 cells (Fig 2C and E), which was also confirmed by immunoblotting (Fig EV3C). Notably, in WT cells, ppMLC and LUZP1 levels within CRs showed a significant strong correlation (Fig 2F), indicating that LUZP1 levels are a main determinant of ppMLC levels within CRs. To further examine the importance of LUZP1 on ppMLC levels within CRs and apical constriction, we examined TJ-deficient ZO-1/-2 DKO cells. In correlation with the apparent reduction of LUZP1 levels within CRs, ppMLC levels within CRs were downregulated (Fig 2G) and apical constriction was disturbed (Fig 2H). Collectively, the findings indicate that LUZP1 at the TJ-associated CRs is crucial for upregulating ppMLC levels within CRs to induce apical constriction. Notably, LUZP1 KO reduced ppMLC levels within CRs and led to apical constriction defects in other epithelial cell lines, such as MTD-1A cells (epithelial cells derived from malignant neoplasms of the mouse mammary gland; Figs 2I and EV2H) and CSG120/7 cells (epithelial cells derived from malignant neoplasms of the mouse submandibular gland; Figs 2J and EV2H), suggesting that LUZP1-mediated upregulation of ppMLC with CRs is a conserved phenomenon across different epithelial cells.

We also found that the effect of LUZP1 KO on ppMLC levels within CRs differed depending on the confluency state of epithelial Eph4 cells. It is known—at least in some types of epithelial cells

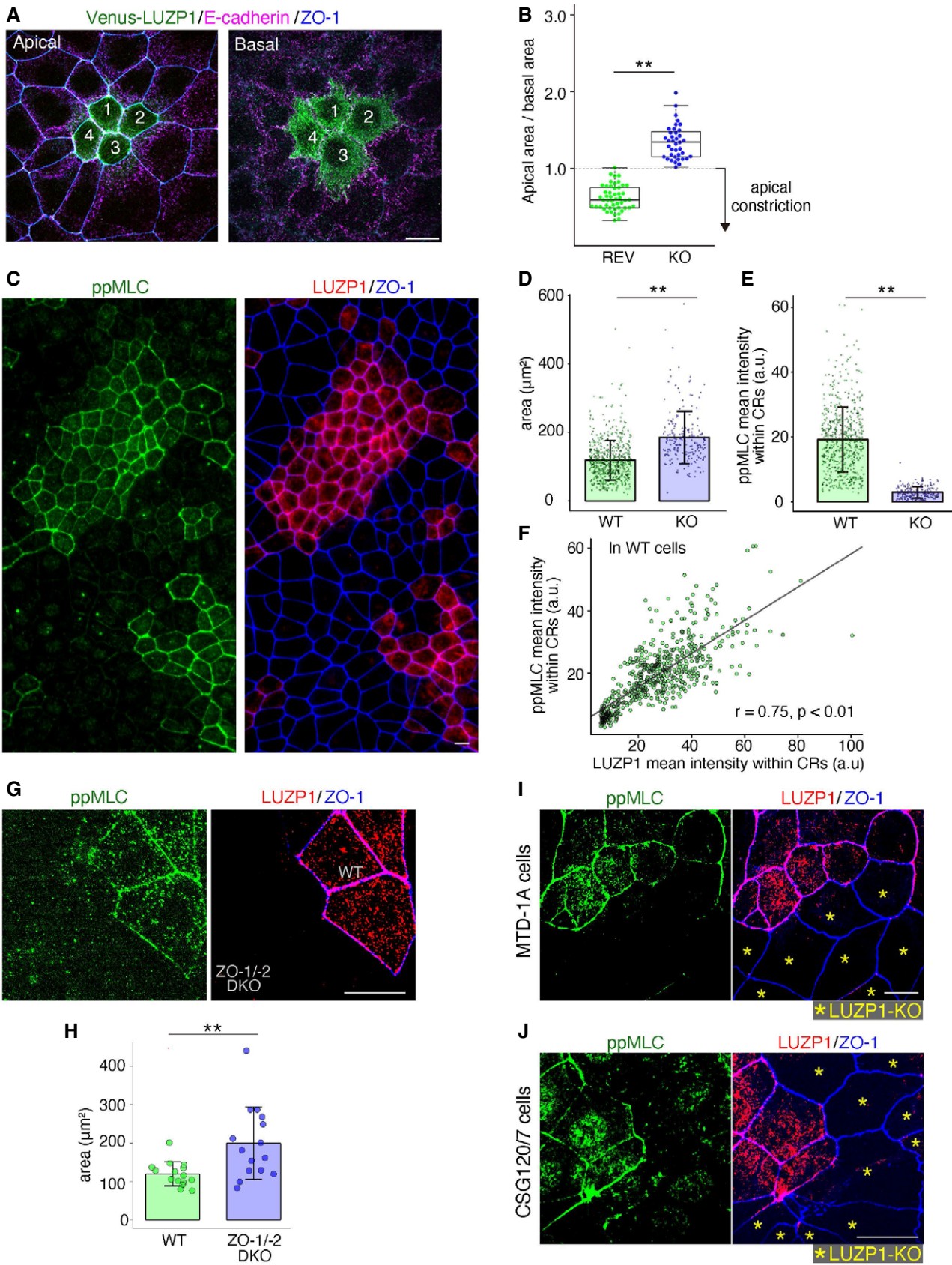

**Figure 2.**

◄ **Figure 2.  LUZP1 knockout (LUZP1 KO) cells display apical constriction defects with a significant reduction in di-phosphorylated myosin light chain (ppMLC) levels within circumferential rings (CRs).**

A   Representative confocal micrographs of co-cultures of Venus-LUZP1-expressing LUZP1 KO (REV) and LUZP1 KO Eph4 cells in the apical and basal plane. Scale bar, 10 μm.

B   Box plots with dot density plots showing that the apical area/basal area ratio was significantly smaller in REV cells than in LUZP1 KO cells (0.65 ± 0.16 [REV] vs. 1.30 ± 0.17 [LUZP1 KO]). *n* = 10. **P* < 0.01 (unpaired *t*-test). The value < 1.0 indicates apical constriction. The solid lines represent the medians, and the boxes represent the interquartile ranges. The error bars extending from the box represent the data within 1.5 times of the interquartile range.

C   Representative micrographs of co-cultures of wild-type (WT) and LUZP1 KO Eph4 cells. ppMLC levels within CRs were clearly reduced in LUZP1 KO cells compared with those in WT cells. Scale bar, 10 μm.

D   Bar plots with dot density plots showing that apical areas of WT cells were significantly smaller than those of LUZP1 KO cells (118.34 ± 57.84 μm² [WT] vs. 185.32 ± 76.69 μm² [LUZP1 KO]). *n* = 3. **P* < 0.01 (Mann–Whitney *U* test). Bars and error bars represent the mean ± standard deviation (SD).

E   Bar plots with dot density plots showing that ppMLC levels within CRs were significantly reduced in LUZP1 KO cells compared with those in WT cells (19.19 ± 10.02 arbitrary units [a.u.] [WT] vs. 2.92 ± 1.81 a.u. [LUZP1 KO]). *n* = 3. **P* < 0.01 (Mann–Whitney *U* test). Bars and error bars represent the mean ± SD.

F   A scatter plot showing that mean fluorescence intensities of ppMLC and those of LUZP1 significantly correlated within CRs (Pearson's correlation coefficients [*r*] = 0.75, *P* < 0.01).

G   Representative confocal micrographs of co-cultures of wild-type (WT) and ZO-1/-2 double knockout (DKO) Eph4 cells. ppMLC levels within CRs were clearly reduced in ZO-1/-2 DKO cells compared with those in WT cells. Scale bar, 10 μm.

H   Bar plots with dot density plots showing that apical areas of WT cells were significantly smaller than those of ZO-1/-2 DKO cells (119.62 ± 31.62 μm² [WT] vs. 199.96 ± 94.22 μm² [LUZP1 KO]; See also Fig 1I). *n* = 3. **P* < 0.01 (unpaired *t*-test). Bars and error bars represent the mean ± SD.

I   Representative confocal micrographs of co-cultures of WT and LUZP1 KO MTD-1A cells. ppMLC levels within CRs were clearly reduced in LUZP1 KO MTD-1A cells compared with those in WT MTD-1A cells. Scale bar, 10 μm.

J   Representative confocal micrographs of co-cultures of WT and LUZP1 KO CSG120/7 cells. ppMLC levels within CRs were clearly reduced in LUZP1 KO CSG120/7 cells compared with those in WT CSG120/7 cells. Scale bar, 10 μm.

such as Eph4 cells and Madin-Darby canine kidney cells—that in a low-confluency state when initial adhesion is made, epithelial cells have immature CRs not associated with myosin II (Kishikawa *et al*, 2008; Furukawa *et al*, 2017) (Fig EV3D). It is also known that even after a primary high-confluency is established with CRs starting to associate with myosin II, the cell number increases until a contact-inhibited high-confluency state, when myosin II is fully integrated into CRs and ppMLC is prominently upregulated (Fig EV3D). Immunofluorescence analyses revealed that in low-confluency and primary high-confluency states, the effect of LUZP1 on ppMLC levels was not evident. However, in a contact-inhibited high-confluency state, the effect of LUZP1 on ppMLC levels was clear (Fig EV3E), suggesting that once ppMLC levels exceed a certain threshold, LUZP1 becomes important for maintaining or further upregulating ppMLC levels.

## LUZP1 binds more strongly to di-phosphomimetic MLC (DD-MLC) than to wild-type MLC (WT-MLC) and di-dephosphomimetic MLC (AA-MLC)

Another important finding is that transient treatment with 100 μM Y27632 (the ROCK inhibitor), which enables us to artificially reduce ppMLC levels without any apparent effect on MLC localization (Fig EV3F), inhibited LUZP1 junctional localization (Figs 3A and B, and EV3G, and Movie EV2). Consistent with this, artificial upregulation of ppMLC levels using calyculin A, an inhibitor of protein phosphatase 1 and protein phosphatase 2A, promoted LUZP1 junctional localization (Fig 3C and D). Altogether, these findings indicate that ppMLC within CRs promotes junctional localization of LUZP1, where LUZP1 upregulates ppMLC levels. Thus, ppMLC and LUZP1 can be considered to create a positive feedback loop that assures the robustness of increased ppMLC levels within CRs to promote apical constriction (Fig 3E).

Our findings thus far prompted two questions; (i) how ppMLC, and not MLC itself, determines LUZP1 junctional localization and (ii) how LUZP1 upregulates ppMLC levels within CRs. To answer

the first question, we hypothesized that MLC binding affinity for LUZP1 depends on its phosphorylation status with ppMLC having a very strong binding affinity because ppMLC and LUZP1 were highly co-localized within CRs (Fig 3F and G). Therefore, we tested this hypothesis using *in vitro* direct binding assays, which revealed that LUZP1 bound more strongly to DD-MLC (with T18D and S19D mutations; $K_d = 0.85 \pm 0.39$ μM) than to WT-MLC ($K_d = 4.04 \pm 1.05$ μM) and AA-MLC (with T18A and S19A mutations; $K_d = 4.88 \pm 1.79$ μM; Fig 3H–K). Altogether, these results indicate that LUZP1 binds more strongly to ppMLC than to other forms of MLC, which would be critical for LUZP1 junctional localization.

## LUZP1 inhibits myosin phosphatase by suppressing the activity of its catalytic subunit, protein phosphatase 1c β/δ (PP1c β/δ)

We next examined how LUZP1 upregulates ppMLC levels within CRs. Because ROCK was reported to be the primary kinase driving MLC phosphorylation, we first analyzed the difference in ROCK1 localization between WT and LUZP1 KO cells. However, ROCK1 fluorescent intensity within CRs as well as that of Shroom3, a well-known interactor of ROCK1, was similar between WT and LUZP1 KO cells (Figs 4A and B, and EV4A). Next, we examined the effects of LUZP1 on ROCK1 function by performing *in vitro* MLC phosphorylation assays using GST-MLC, GST-ROCK1 catalytic domain, and GST-LUZP1. No evidence was found that LUZP1 affects ROCK1 function (Fig 4C).

We then hypothesized that LUZP1 upregulates ppMLC levels via its effect on myosin phosphatase. Supporting this hypothesis, treatment with calyculin A reversed the difference in ppMLC levels within CRs between REV and LUZP1 KO cells, as evidenced by immunofluorescence analyses (Fig 4D and E) and immunoblotting (Fig 4F and G). To further pursue this possibility, we first compared the localization of PP1c β/δ, which is the catalytic subunit of myosin phosphatase (Fig 5A), between WT and LUZP1 KO cells. However, PP1c localization was similar (Fig 5B and C). We next examined the

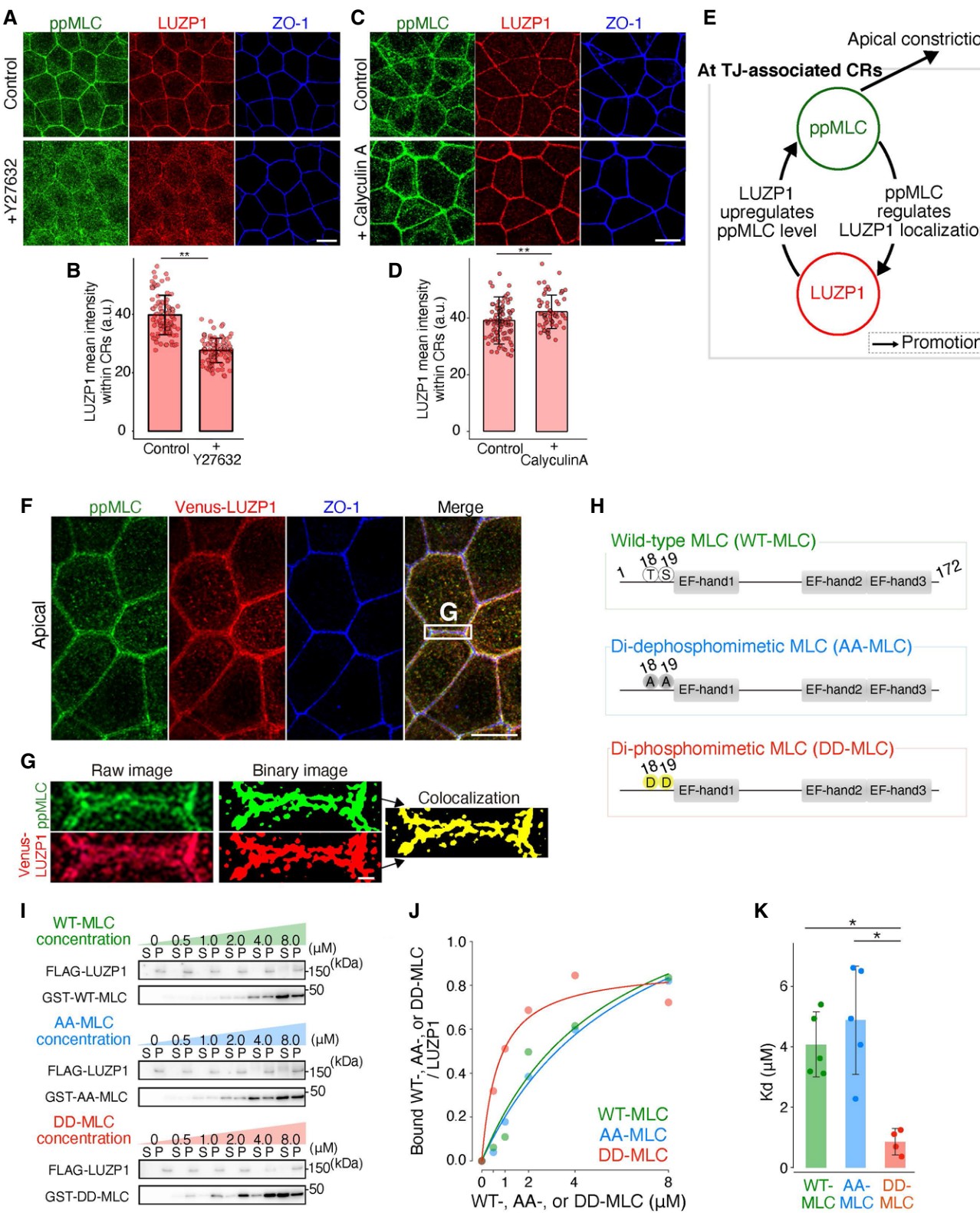

Figure 3.

**Figure 3.   LUZP1 binding affinity for myosin light chain (MLC) depends on the phosphorylation status of MLC.**

A   Representative confocal micrographs of wild-type (WT) Eph4 cells treated with 100 μM Y27632 for 30 min. Artificial reductions in di-phosphorylated myosin light chain (ppMLC) levels with Y27632 inhibited LUZP1 junctional localization. Scale bar, 10 μm.

B   Bar plots with dot density plots showing that LUZP1 mean intensities within circumferential rings (CRs) were significantly reduced after Y27632 treatment (39.75 ± 6.74 arbitrary units [a.u.] [control] vs. 27.7 ± 4.21 a.u. [Y27632]). $n = 3$. **$P < 0.01$ (Mann–Whitney $U$ test). Bars and error bars represent the mean ± standard deviation (SD).

C   Representative confocal micrographs of WT Eph4 cells treated with 100 nM Calyculin A for 30 min. Artificial upregulations in ppMLC levels with calyculin A promoted LUZP1 junctional localization. Scale bar, 10 μm.

D   Bar plots with dot density plots showing that LUZP1 mean intensities within CRs were significantly upregulated after calyculin A treatment (39.19 ± 8.32 a.u. [control] vs. 42.24 ± 5.40 a.u. [calyculin A]). $n = 3$. **$P < 0.01$ (Mann–Whitney $U$ test). Bars and error bars represent the mean ± SD.

E   A schematic drawing of the positive feedback loop between ppMLC and LUZP1 at tight junction (TJ)-associated CRs to promote apical constriction.

F   Representative micrographs of Venus-LUZP1-expressing LUZP1 KO (REV) Eph4 cells in the apical plane. Scale bar, 10 μm.

G   Magnified micrographs of Fig 3F. Binary images on the right show the co-localization of ppMLC and LUZP1. Scale bar, 1 μm.

H   Schematics of wild-type MLC (WT-MLC), di-dephosphomimetic MLC (AA-MLC, with T18A and S19A mutations), and di-phosphomimetic MLC (DD-MLC, with T18D and S19D mutations).

I   *In vitro* direct binding assay between FLAG-LUZP1 and GST-MLC. Representative immunoblots are shown. S, supernatant; P, pellet.

J   A representative plot in which the amount of bound GST-WT-, AA-, or DD-MLC/FLAG-LUZP1 was plotted with the Michaelis–Menten fitted curve.

K   Bar plots with dot density plots showing that the dissociation constant ($K_d$) of LUZP1 from DD-MLC was significantly lower than that from WT-MLC and AA-MLC (4.04 ± 1.05 μM [WT-MLC] vs. 4.88 ± 1.79 μM [AA-MLC] vs. 0.85 ± 0.39 μM [DD-MLC]). $n = 5$ (WT-MLC and AA-MLC) and $n = 4$ (DD-MLC). *$P < 0.05$ (Kruskal–Wallis test followed by Steel–Dwass test). Bars and error bars represent the mean ± SD.

Source data are available online for this figure.

effect of LUZP1 on myosin phosphatase function. After LUZP1 was found to bind PP1c β/δ via co-immunoprecipitation (Fig 5D), we conducted *in vitro* MLC phosphorylation assays, this time using GST-PP1c β/δ in addition to GST-MLC, GST-ROCK1 catalytic domain, and GST-LUZP1. The assays revealed that ppMLC levels gradually increased as LUZP1 levels increased (Fig 5E). Given that LUZP1 failed to increase ppMLC levels in previous *in vitro* MLC phosphorylation assays without GST-PP1c β/δ (Fig 4C), this result strongly indicates that LUZP1 inhibits PP1c β/δ-driven ppMLC dephosphorylation.

Next, we asked whether LUZP1-mediated inhibition of PP1c β/δ-driven ppMLC dephosphorylation relies on (i) competitive inhibition of binding between PP1c β/δ and ppMLC or (ii) inhibition of PP1c β/δ activity. To answer this question, we conducted further *in vitro* phosphorylation assays using a PP1c β/δ substrate without binding affinity for LUZP1. Merlin is phosphorylated by p21-activated kinase 1 (PAK1) to phosphorylated Merlin (pMerlin) (Xiao *et al*, 2002; Ye, 2007), which can be dephosphorylated by PP1c β/δ (Morrison *et al*, 2001; Jin *et al*, 2006). Pull-down assays revealed that Merlin did not bind LUZP1 (Fig EV4B). Therefore, we conducted *in vitro* Merlin phosphorylation assays using GST-PP1c β/δ, GST-Merlin, PAK1, and GST-LUZP1 and found that pMerlin levels increased in the presence of LUZP1, indicating that LUZP1 inhibits the activity of PP1c β/δ (Fig 5F). Altogether, our findings indicate that LUZP1 inhibits myosin phosphatase by suppressing the activity of its catalytic subunit, PP1c β/δ (Fig 5G).

**MTs promote LUZP1-mediated inhibition of myosin phosphatase, thereby upregulating ppMLC levels within CRs to promote apical constriction**

Thus far, we revealed that LUZP1 is recruited to TJ-associated CRs via its strong binding affinity for ppMLC when TJs are intact, inhibiting myosin phosphatase to maintain upregulated ppMLC levels within CRs for vertebrate apical constriction. Because we identified LUZP1 as an MT-binding protein at first, we next examined the bidirectional influence of MTs and LUZP1. Immunofluorescence analyses revealed that LUZP1 had no influence on apical MT

organization (Fig EV4C and D). However, treatment with nocodazole, a potent MT-depolymerizing agent, reversed the apical constriction of REV cells in co-cultures of REV and LUZP1 KO cells (Figs 6A and EV4E). Consistent with this, in co-cultures of WT and LUZP1 KO cells, nocodazole reduced ppMLC levels within CRs only in WT cells, whereas no effect was observed in LUZP1 KO cells (Fig 6B and C). These results suggest that the effect of MTs on apical constriction and ppMLC levels depends on LUZP1. Next, we theorized regarding the possible mechanisms behind this phenomenon and considered that (i) MTs may alter the binding affinity between ppMLC and LUZP1, thereby promoting LUZP1 junctional localization and/or (ii) MTs facilitate LUZP1-mediated inhibition of myosin phosphatase. We rejected the first possibility because the degree of co-localization between ppMLC and LUZP1 did not differ after nocodazole treatment (Fig EV4F). In contrast, *in vitro* MLC phosphorylation assays, using MTs in addition to GST-MLC, GST-ROCK1 catalytic domain, GST-PP1c β/δ, and GST-LUZP1 revealed that MTs promoted MLC phosphorylation only in the presence of LUZP1 (Fig 6D). Similarly, *in vitro* Merlin phosphorylation assays using MTs in addition to GST-Merlin, PAK1, GST-PP1c β/δ, and GST-LUZP1 showed that MTs upregulated Merlin phosphorylation only in the presence of LUZP1 (Fig EV4G). Based on these results, we concluded that LUZP1-mediated inhibition of myosin phosphatase is facilitated by MTs, elucidating the previously unexplained link between MTs, LUZP1, and vertebrate apical constriction (Fig 6E).

## Discussion

Here, we revealed that ppMLC promotes the recruitment of LUZP1 to TJ-associated CRs where LUZP1 spatiotemporally inhibits myosin phosphatase in an MT-facilitated manner for vertebrate apical constriction. This finding contributes to our understanding of the regulatory mechanism of vertebrate apical constriction in several respects (Fig 7). First, LUZP1-mediated inhibition of myosin phosphatase represents a novel way of promoting MLC phosphorylation status within CRs to promote apical constriction. Second, it is notable that ppMLC itself spatiotemporally triggers this LUZP1-

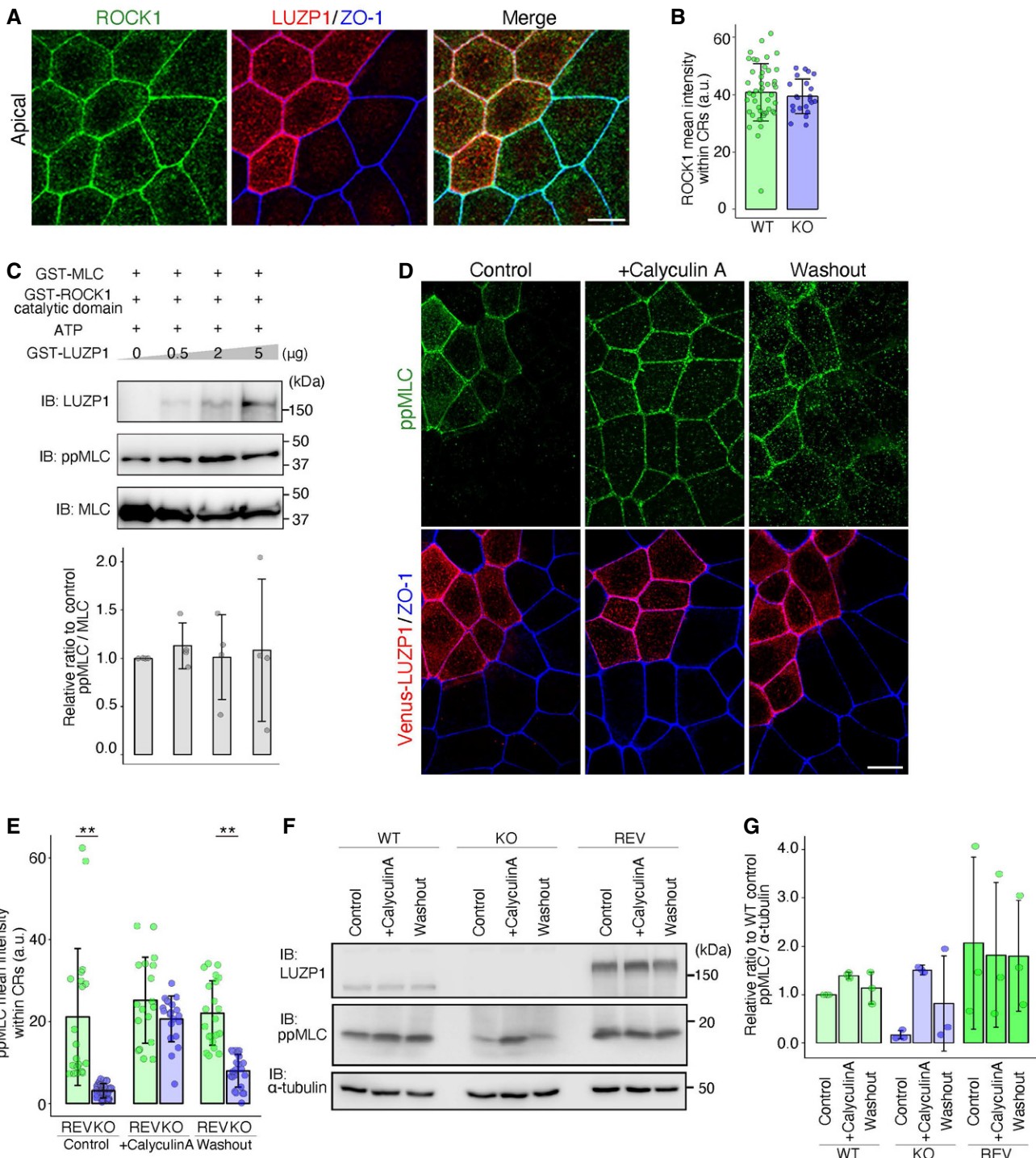

**Figure 4.**

Figure 4. Calyculin A treatment reverses the effect of LUZP1 on di-phosphorylated myosin light chain (ppMLC) levels.

A   Representative confocal micrographs of co-cultures of wild-type (WT) and LUZP1 knockout (LUZP1 KO) Eph4 cells in the apical plane. Scale bar, 10 μm.
B   Bar plots with dot density plots showing that ROCK1 mean intensities within circumferential rings (CRs) are similar between WT and LUZP1 KO cells (40.87 ± 9.95 arbitrary units [a.u.] [WT] vs. 39.48 ± 6.04 a.u. [LUZP1 KO]). $n = 3$. $P = 0.54$ (Mann–Whitney $U$ test). Bars and error bars represent the mean ± standard deviation (SD).
C   *In vitro* myosin light chain (MLC) phosphorylation assay using 25 ng GST-MLC, 4 ng GST-ROCK1 catalytic domain, 1 mM ATP, and 0–5 μg GST-LUZP1. Quantification of the ppMLC/MLC ratio relative to the control showed that LUZP1 did not change the ratio (1.00 [1st lane, control] vs. 1.13 ± 0.24 [2nd lane] vs. 1.01 ± 0.44 [3rd lane] vs. 1.08 ± 0.73 [4th lane]). $n = 4$. $P = 0.49$ (Kruskal–Wallis test). Bars and error bars represent the mean ± SD. IB, immunoblotting.
D   Representative confocal micrographs of co-cultures of Venus-LUZP1-expressing LUZP1 KO (REV) and LUZP1 KO Eph4 cells treated with 100 nM calyculin A for 30 min. Scale bar, 10 μm.
E   Bar plots with dot density plots showing that calyculin A reversed the difference in ppMLC levels within CRs between REV and LUZP1 KO cells (control, 21.14 ± 16.80 a.u. [WT] vs. 3.10 ± 1.72 a.u. [LUZP1 KO]; calyculin A, 25.24 ± 10.54 a.u. [WT] vs. 20.65 ± 5.62 a.u. [LUZP1 KO]; washout, 22.09 ± 7.90 a.u. [WT] vs. 7.92 ± 4.01 a.u. [LUZP1 KO]). **$P < 0.01$ (Mann–Whitney $U$ test). Bars and error bars represent the mean ± SD. $n = 3$.
F   Representative immunoblot of WT, LUZP1 KO, and Venus-LUZP1-expressing LUZP1 knockout (REV) Eph4 cells treated with 100 nM calyculin A for 30 min.
G   Quantification of the ppMLC/MLC ratio relative to WT control, confirming the reversal of the difference in ppMLC levels within CRs between WT and LUZP1 KO cells by calyculin A (WT, 1.00 [control] vs. 1.40 ± 0.06 [calyculin A] vs. 1.14 ± 0.33 [washout]; KO, 0.09 ± 0.04 [control] vs. 1.49 ± 0.06 [calyculin A] vs. 0.81 ± 0.99 [washout]; REV, 2.06 ± 1.78 [control] vs. 1.82 ± 1.50 [calyculin A] vs. 1.80 ± 1.14 [washout]). $n = 3$. Bars and error bars represent the mean ± SD.

Source data are available online for this figure.

based system to protect itself. Third, the LUZP1-based system not only explains the mechanism behind the indispensable role of MTs in vertebrate apical constriction for the first time but also represents a novel actomyosin–MT crosstalk mechanism. Finally, this system is also unique in that it functions primarily around TJs, not AJs, in AJCs, for apical constriction.

In vertebrates, it is known that AJC-localized ROCK is the primary driving kinase that upregulates ppMLC levels within CRs to induce apical constriction (Sawyer *et al*, 2010; Suzuki *et al*, 2012; Martin & Goldstein, 2014; Takeichi, 2014). Our study demonstrated that after ppMLC levels are upregulated, ppMLC recruits LUZP1 to TJ-associated CRs and LUZP1 spatiotemporally inhibits myosin phosphatase in an MT-facilitated manner, maintaining increased ppMLC levels within CRs. Therefore, in synergy with AJC-localized ROCK, the MT-facilitated LUZP1-mediated inhibition of myosin phosphatase can be considered as a system that assures the robustness of increased ppMLC levels within CRs to further promote apical constriction. This explains the previous identification of LUZP1 as an NTD gene (Hsu *et al*, 2008). Early studies identified LUZP1 as a protein almost exclusively expressed in the brain (Sun *et al*, 1996), but more recent studies demonstrated ubiquitous expression of LUZP1 mRNA throughout other tissues (Ono *et al*, 2017) (https://refex.dbcls.jp), which is consistent with our data (Fig EV1H). Most recently, it was reported that LUZP1 affects actin organization due to its binding affinity for filamin A (Wang & Nakamura, 2019). However, our study demonstrated that in cultured epithelial Eph4 cells, LUZP1 does not affect actin organization but promotes contraction of actomyosin filaments by inhibiting PP1c β/δ activity. Most notably, a recent non-biased interactome database showed that LUZP1 interacts with PP1c (Hein *et al*, 2015) (https://thebiogrid.org), strongly supporting our observation.

Our results also uncovered a novel mechanism connecting MTs to the actomyosin network (Dogterom & Koenderink, 2019). Interestingly, previous studies have suggested that MTs can context-dependently regulate actomyosin contraction both positively and negatively. For example, MTs can promote actomyosin contraction by enhancing actin assembly (Svitkina *et al*, 2003; Lewkowicz *et al*, 2008; Henty-Ridilla *et al*, 2016). Conversely, MTs can sequester and inhibit GEF-H1, a facilitator of the RhoA-ROCK pathway, thereby locally inhibiting actomyosin contraction (Krendel *et al*, 2002; Nagae *et al*, 2013; Rafiq *et al*, 2019). MTs can also provide a pushing

force to resist actomyosin contraction (Singh *et al*, 2018; Takeda *et al*, 2018). However, regarding apical constriction, MTs have consistently been shown to promote actomyosin contraction (Lee & Harland, 2007; Booth *et al*, 2014; Fernandes *et al*, 2014; Ko *et al*, 2019). In the invertebrate *Drosophila*, proper salivary gland tubulogenesis requires MTs as they facilitate the formation of a medioapical actomyosin network (Booth *et al*, 2014), and proper mesoderm cell invagination requires MTs to help connect the medioapical actomyosin network to AJs (Ko *et al*, 2019). Our finding that MTs can contribute to the inhibition of myosin phosphatase via LUZP1 constitutes another novel mechanism explaining the dependence of apical constriction on MTs and the first for vertebrate apical constriction. How apical MTs elaborately coordinate these different mechanisms for proper apical constriction is an important issue that warrants future investigation.

Finally, this LUZP1-based mechanism sheds new light on the importance of the association between TJs and CRs for vertebrate apical constriction. TJs are vertebrate-specific junctions that are not only crucial for paracellular barriers with selective paracellular permeability, but also as signaling hubs regulating a variety of cellular events (Tsukita *et al*, 2001; Raleigh *et al*, 2011; Krug *et al*, 2014; Suzuki *et al*, 2014; Tamura & Tsukita, 2014; Saitoh *et al*, 2015; Tanaka *et al*, 2015, 2016; Zihni *et al*, 2016; Tanaka *et al*, 2017; Odenwald *et al*, 2018; Tanaka *et al*, 2018; Citi, 2019; Nakamura *et al*, 2019; Shigetomi & Ikenouchi, 2019; Tsukita *et al*, 2019b). Notably, in the context of TJ maturation, the association between actomyosin CRs and TJs is known to be important. For example, pharmacological alternation of CRs leads to the disruption of TJs (Rodgers & Fanning, 2011), and localized RhoA activity induced by two Rho GEFs, namely ARHGEF11 and p114 RhoGEF, is required for proper TJ maturation (Terry *et al*, 2011; Itoh *et al*, 2012; Zihni & Terry, 2015). However, in the context of apical constriction regulation, the association between actomyosin CRs and AJs has attracted more attention than that between CRs and TJs, partly because the invertebrate *Drosophila*, the most popular model organism for studying apical constriction, possesses only AJs and not TJs. Considering that this ppMLC-triggered, LUZP1-based system specifically works at TJ-associated CRs and not at AJ-associated CRs, our study implies that TJ- and AJ-associated CRs may have some distinct roles, warranting further studies focusing on the different roles between TJ- and AJ-

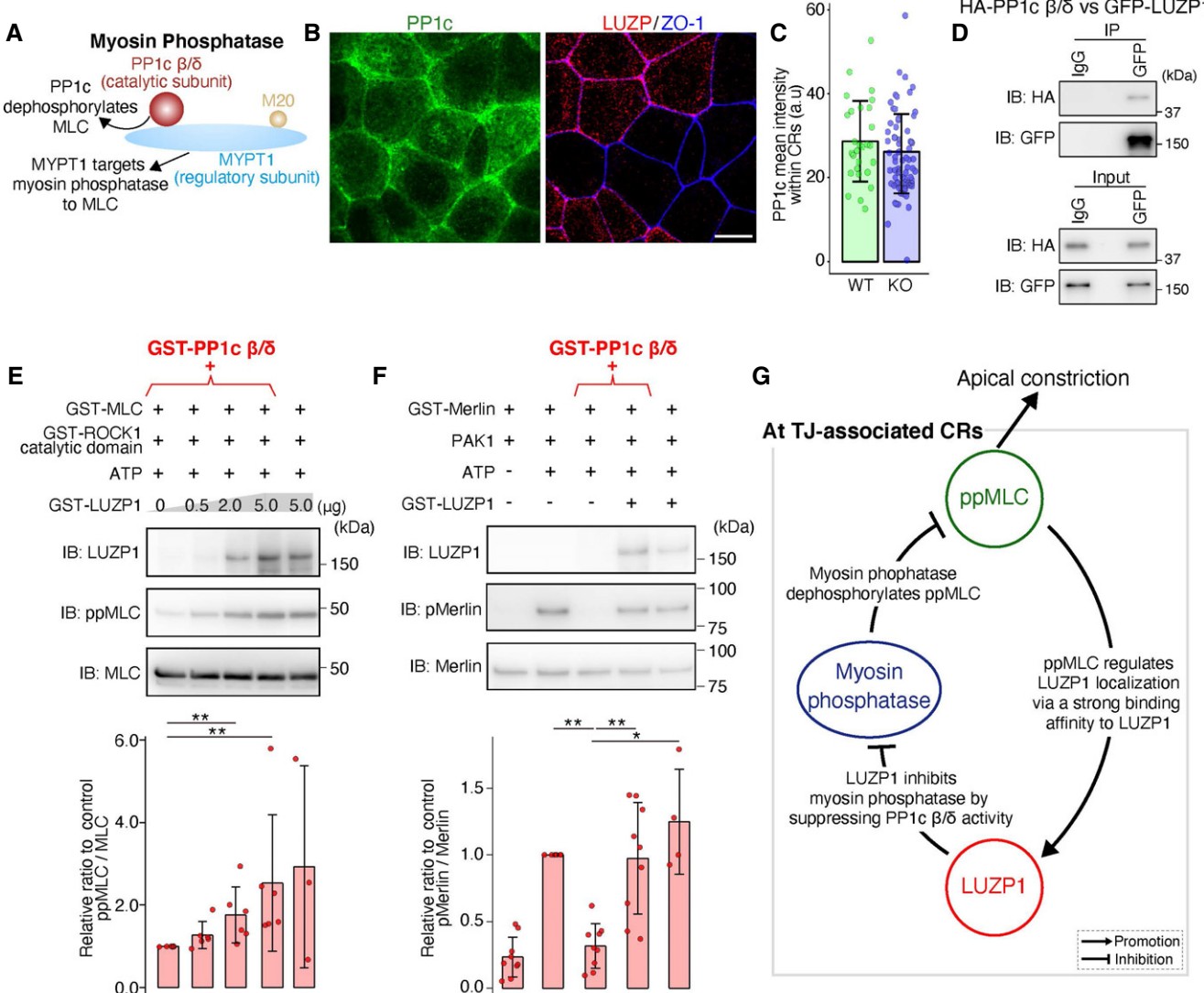

**Figure 5. LUZP1 inhibits the activity of protein phosphatase 1c β/δ (PP1c β/δ), the catalytic subunit of myosin phosphatase.**

A   A schematic drawing of myosin phosphatase. Myosin phosphatase consists of PP1c β/δ, myosin phosphatase targeting subunit 1 (MYPT1), and a small 20-kDa regulatory subunit (M20). PP1c β/δ represents a catalytic subunit responsible for dephosphorylating myosin light chain (MLC), whereas MYPT1 targets myosin phosphatase to MLC by binding both PP1c β/δ and MLC.

B   Representative confocal micrographs of co-cultures of wild-type (WT) and LUZP1 knockout (LUZP1 KO) Eph4 cells in the apical plane. Scale bar, 10 μm.

C   Bar plots with dot density plots showing that PP1c mean intensities within CRs are similar between WT and LUZP1 KO cells ($28.68 \pm 9.60$ arbitrary units [a.u.] [WT] vs. $25.04 \pm 9.47$ a.u. [LUZP1 KO]). $P = 0.09$ [Mann–Whitney $U$ test]. $n = 3$. Bars and error bars represent the mean $\pm$ standard deviation (SD).

D   Co-immunoprecipitation of HA-PP1c β/δ and GFP-LUZP1. LUZP1 binds to PP1c β/δ. IB, immunoblotting.

E   *In vitro* MLC phosphorylation assay using 1 μg GST-PP1c β/δ in addition to 25 ng GST-MLC, 4 ng GST-ROCK1 catalytic domain, 1 mM ATP, and 0–5 μg GST-LUZP1. Quantification of the di-phosphorylated MLC (ppMLC)/MLC ratio relative to the control showed that LUZP1 upregulated ppMLC/MLC levels in a dose-dependent manner (1.00 [1st lane, control] vs. $1.27 \pm 0.33$ [2nd lane] vs. $1.76 \pm 0.68$ [3rd lane] vs. $2.53 \pm 1.65$ [4th lane] vs. $2.93 \pm 2.45$ [5th lane]). $n = 3$ or 6. **$P < 0.01$ (Kruskal–Wallis test followed by Steel test [compared with 1st lane]). Bars and error bars represent the mean $\pm$ SD.

F   *In vitro* Merlin phosphorylation assay using 1 μg GST-PP1c β/δ, 100 ng GST-Merlin, 2 pg p21-activated kinase 1 (PAK1), and 5 μg GST-LUZP1. Quantification of the phosphorylated Merlin (pMerlin)/Merlin ratio relative to the control showed that LUZP1 upregulated pMerlin/Merlin levels ($0.23 \pm 0.15$ [1st lane] vs. 1.00 [2nd lane, control] vs. $0.32 \pm 0.17$ [3rd lane] vs. $0.97 \pm 0.42$ [4th lane] vs. $1.25 \pm 0.39$ [5th lane]). $n = 4$ or 9. *$P < 0.05$, **$P < 0.01$ (Kruskal–Wallis test followed by Steel test [compared with 3rd lane]). Bars and error bars represent the mean $\pm$ SD.

G   A schematic drawing of the relationships among ppMLC, LUZP1, and myosin phosphatase at tight junction (TJ)-associated CRs to promote apical constriction.

Source data are available online for this figure.

associated CRs. Hopefully, our findings lead to further elucidation of the significance of TJ and CR association in epithelial morphogenesis.

In summary, we revealed that ppMLC recruits LUZP1 to TJ-associated CRs where LUZP1 spatiotemporally inhibits myosin phosphatase in an MT-facilitated manner. This ppMLC-triggered, MT-

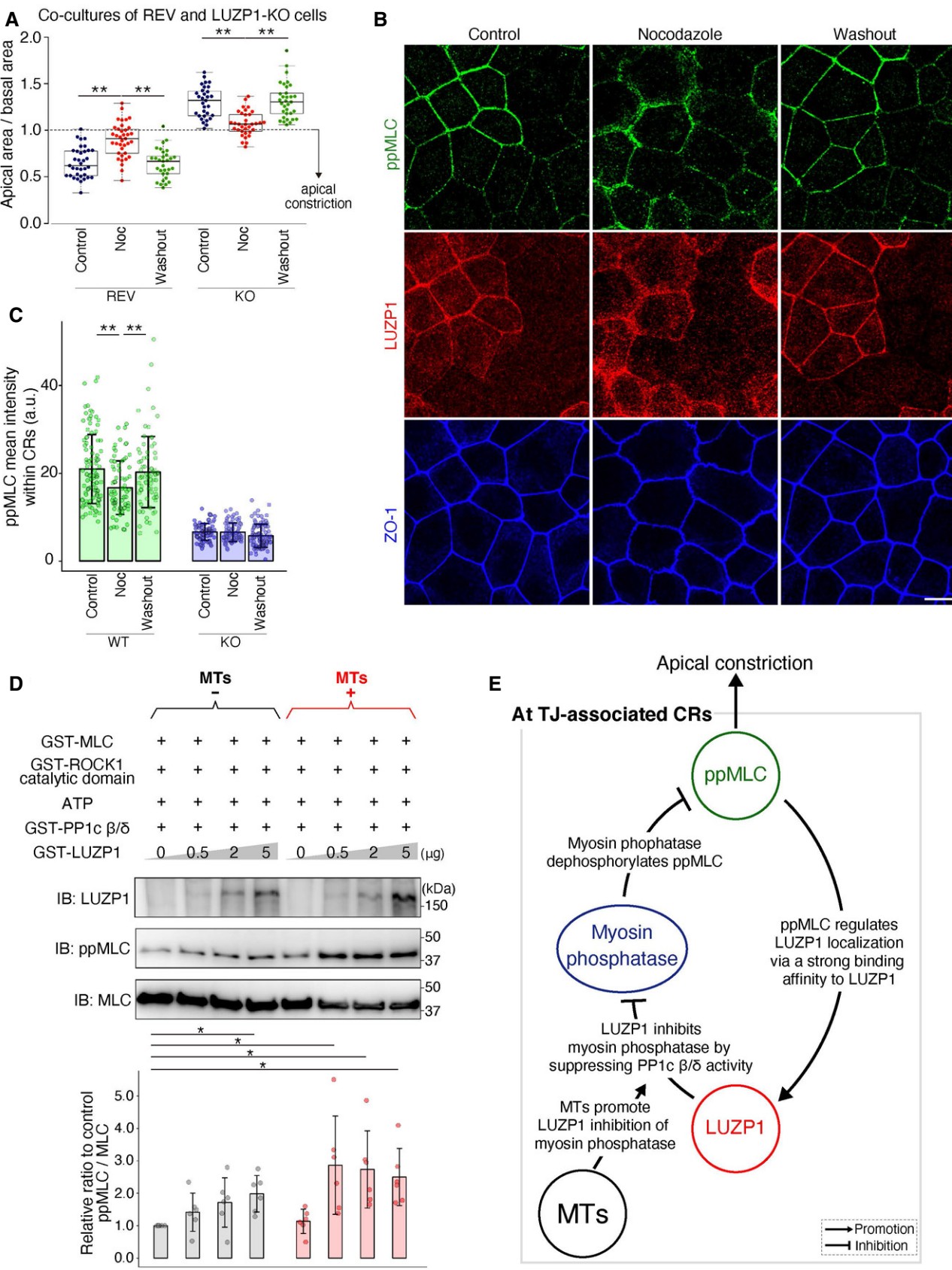

Figure 6.

**Figure 6.  Microtubules (MTs) promote LUZP1 inhibition of myosin phosphatase.**

A  Box plots with dot density plots showing the ratio of the apical area/basal area in co-cultures of Venus-LUZP1-expressing LUZP1 knockout (REV) and LUZP1 knockout (LUZP1 KO) Eph4 cells; 2 μM nocodazole treatment for 30 min partially reversed apical constriction of REV cells (REV, 0.65 ± 0.16 [control] vs. 0.90 ± 0.18 [nocodazole] vs. 0.64 ± 0.16 [washout]; KO, 1.30 ± 0.17 [control] vs. 1.07 ± 0.13 [nocodazole] vs. 1.32 ± 0.19 [washout]). **$P < 0.01$ (Kruskal–Wallis test followed by Steel–Dwass test). The solid lines represent the medians, and the boxes represent the interquartile ranges. The error bars extending from the box represent the data within 1.5 times of the interquartile range.

B  Representative confocal micrographs of co-cultures of LUZP1-expressing wild-type (WT) and LUZP1 KO Eph4 cell treated with 2 μM nocodazole for 30 min. Nocodazole treatment partially reversed the difference in di-phosphorylated MLC (ppMLC) levels within circumferential rings (CRs) between WT and LUZP1 KO cells. Scale bar, 10 μm.

C  Bar plots with dot density plots showing that ppMLC levels within CRs were significantly downregulated in WT Eph4 cells after nocodazole treatment. Importantly, ppMLC levels in LUZP1 KO Eph4 cells were unchanged after nocodazole treatment (WT, 21.43 ± 6.96 arbitrary units [a.u.] [control] vs. 17.67 ± 5.40 a.u. [nocodazole] vs. 20.84 ± 7.19 a.u. [washout]; KO, 8.74 ± 1.71 a.u. [control] vs. 8.67 ± 1.89 a.u. [nocodazole] vs. 7.96 ± 2.35 a.u. [washout]). $n = 3$. **$P < 0.01$ (Kruskal–Wallis test followed by Steel–Dwass test). Bars and error bars represent the mean ± standard deviation (SD).

D  *In vitro* MLC phosphorylation assay using 1 μg MTs in addition to 25 ng GST-MLC, 4 ng GST-ROCK1 catalytic domain, 1 mM ATP, 1 μg GST-protein phosphatase 1c β/δ (PP1c β/δ), and 0–5 μg GST-LUZP1. Quantification of the relative ppMLC/MLC ratio to the control showed that MTs promote LUZP1-mediated inhibition of PP1c β/δ (1.00 [1st-lane, control] vs. 1.42 ± 0.59 [2nd-lane] vs. 1.72 ± 0.76 [3rd-lane] vs. 1.99 ± 0.56 [4th-lane] vs. 1.14 ± 0.37 [5th-lane] vs. 2.87 ± 1.51 [6th-lane] vs. 2.74 ± 1.19 [7th-lane] vs. 2.50 ± 0.88 [8th-lane]). $n = 6$. *$P < 0.05$ (Kruskal–Wallis test followed by Steel test [compared with 1st lane]). Bars and error bars represent the mean ± SD.

E  A schematic drawing of the relationships among MTs, ppMLC, LUZP1, and myosin phosphatase at TJ-associated CRs to promote apical constriction.

Source data are available online for this figure.

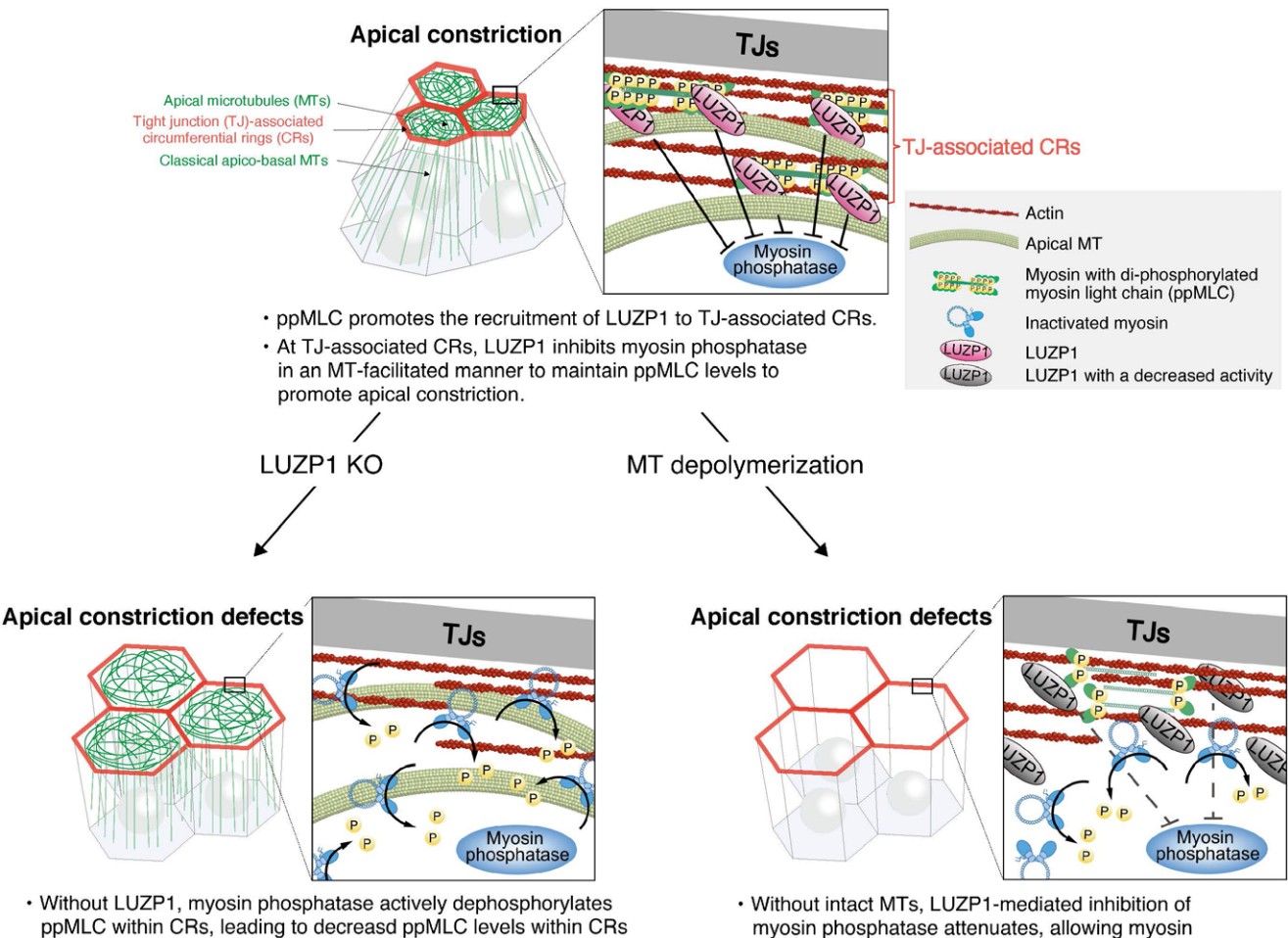

**Figure 7.   A schematic summary of the findings.**
Under normal conditions, ppMLC promotes the recruitment of LUZP1 to TJ-associated CRs, where LUZP1 inhibits myosin phosphatase in a MT-facilitated manner to upregulate ppMLC levels. Without LUZP1, myosin phosphatase actively dephosphorylates ppMLC within CRs, leading to apical constriction defects. MT depolymerization causes attenuated LUZP1-mediated inhibition of myosin phosphatase, allowing myosin phosphatase to dephosphorylate ppMLC.

facilitated, and LUZP1-based mechanism assures the robustness of ppMLC to promote vertebrate apical constriction. We believe that continued research in this direction can eventually lead to a detailed understanding of the regulatory mechanism of apical constriction and the development of treatment strategies for diseases associated with apical constriction defects.

# Materials and Methods

## Reagents and Tools table

| Reagent/Resource | Reference or Source | Identifier or Catalog Number |
|---|---|---|
| **Experimental Models** | | |
| Eph4 cells (*Mus musculus*) | Kindly gifted by Dr. Reichmann (University Children's Hospital Zurich, Zurich, Switzerland) | N/A |
| CSG120/7 cells (*M. musculus*) | Kindly gifted by Dr. Birchmeier (Max-Delbruck-Center for Molecular Medicine, Berlin, Germany) | N/A |
| MTD-1A cells (*M. musculus*) | Kindly gifted by Dr. Takeichi (Riken BDR, Kobe, Japan) | N/A |
| HEK-293 cells (*Homo sapiens*) | ATCC | CRL-1573™ |
| Sf9 cells (*Spodoptera frugiperda*) | Kindly gifted by Drs. Nakamura and Fujiyoshi (Tokyo Medical and Dental University, Tokyo, Japan) | N/A |
| Chick | Local farm | N/A |
| C57BL/6J mice | Japan SLC | http://www.jslc.co.jp/ |
| **Recombinant DNA** | | |
| pCAGGS Venus LUZP1 full-length | This study | N/A |
| pCAGGS FLAG LUZP1 full-length | This study | N/A |
| pCAGGS HA LUZP1 full-length | This study | N/A |
| pCAGGS Venus LUZP1 N (1–353) | This study | N/A |
| pCAGGS Venus LUZP1 M (354–706) | This study | N/A |
| pCAGGS Venus LUZP1 C (707–1,068) | This study | N/A |
| pCAGGS HA LUZP1 N (1–353) | This study | N/A |
| pCAGGS HA LUZP1 M (354–706) | This study | N/A |
| pCAGGS HA LUZP1 C (707–1,068) | This study | N/A |
| pCAGGS HA PP1cβ/δ | This study | N/A |
| pCXN2-HaloTag | This study | N/A |
| pCXN2-HaloTag LUZP1 | This study | N/A |
| pCAGGS GFP α-Tubulin | This study | N/A |
| pCAGGS HA MLC | This study | N/A |
| pCXN2-paGFP | This study | N/A |
| pCXN2-paGFP WT-MLC | This study | N/A |
| pCXN2-paGFP DD-MLC | This study | N/A |
| pCXN2-paGFP AA-MLC | This study | N/A |
| pX330-U6-Chimeric_BB-CBh-hSpCas9 | Cong *et al* (2013) | Addgene plasmid#42230 |
| pGEX6P2 vector | GE Healthcare | #28-9546-50 |
| pGEX LUZP1 full-length | This study | N/A |
| pGEX LUZP1 LUZP1 N (1–353) | This study | N/A |
| pGEX LUZP1 LUZP1 M (354–706) | This study | N/A |
| pGEX LUZP1 LUZP1 C (707–1,068) | This study | N/A |
| pGEX WT MLC | This study | N/A |
| pGEX DD MLC | This study | N/A |
| pGEX AA MLC | This study | N/A |

**Reagents and Tools table**  (continued)

| Reagent/Resource | Reference or Source | Identifier or Catalog Number |
|---|---|---|
| pGEX PP1cβ/δ | This study | N/A |
| pGEX Merlin | This study | N/A |
| p330x LUZP1 KO 1 | This study | N/A |
| p330x LUZP1 KO 2 | This study | N/A |
| p330x LUZP1 KO 3 | This study | N/A |
| p330x ZO-2 KO | This study | N/A |
| p330x E-cadherin KO | This study | N/A |
| **Antibodies** | | |
| Rabbit anti-myosin light chain 2 pAb, IF (1:200), WB (1:500) | Cell Signaling Technology | Cat# 672; RRID: AB_10692513 |
| Rabbit anti-phospho-myosin light chain 2 (Thr18/Ser19) pAb, IF (1:200), WB (1:500) | Cell Signaling Technology | Cat#3671; RRID: AB_330248 |
| Rabit anti-myosin heavy chain B pAb, IF (1:200) | Covance | Cat#PRB-445P-100; RRID: AB_291639 |
| Rabbit anti-merlin pAb, WB (1:500) | Cell Signaling Technology | Cat#12888; RRID: AB_2650551 |
| Rabbit anti-phospho-merlin (Ser518), WB (1:500) | Cell Signaling Technology | Cat#9163; RRID: AB_2149793 |
| Rabbit anti-ZO-1 pAb, IF (1:200) | Life Technologies | Cat#61-7300; RRID: AB_2533938 |
| Goat anti-ZO-2 pAb, IF (1:200) | Santa Cruz | Cat#sc-8148; RRID: AB_2271821 |
| Mouse anti-α-tubulin mAb, IF (1:200), WB (1:500) | Sigma-Aldrich | Cat#T9026; RRID: AB_477593 |
| Rat anti-tubulin mAb, WB (1:500) | Abcam | Cat#ab6160; RRID: AB_305328 |
| Rabbit anti-β-catenin pAb, IEM (1:200) | Sigma-Aldrich | Cat#C2206; RRID: AB_476831 |
| Rabbit anti-claudin-7 pAb, IEM (1:200) | Invitrogen | Cat#34-1700; RRID: AB_2533158 |
| Mouse anti-GFP mAb, WB (1:500) | Life Technologies | Cat#34-9100; RRID: AB_2533190 |
| Rat anti-GFP mAb, WB (1:500) | Nacalai Tesque | Cat#GF090R; RRID: AB_2314545 |
| Mouse anti-HA mAb, WB (1:500) | Covance | Cat#MMS-101R; RRID: AB_291262 |
| Rat anti-HA mAb, WB (1:500) | Roche | Cat#11867431001; RRID: AB_390919 |
| Rabbit anti-PP1c mAb, IF (1:200), WB (1:500) | Abcam | Cat#ab53315; RRID: AB_2168274 |
| Mouse anti-FLAG mAb, WB (1:500) | Sigma-Aldrich | Cat#F1804; RRID: AB_262044 |
| Rat anti-FLAG mAb, WB (1:500) | Novus | Cat#NBP1-06712; RRID: AB_1625981 |
| Rat anti-E-cadherin mAb, IF (Undiluted), IEM (Undiluted) | Kindly gifted by Dr. Takeichi (Riken BDR, Kobe, Japan) | N/A |
| Rat anti-ROCK1 pAb IF (1:100) | Nishimura and Takeichi (2008) | N/A |
| Rabbit anti-Shroom3 pAb IF (1:100) | Nishimura and Takeichi (2008) | N/A |
| Mouse anti-cingulin, IF (Undiluted), IEM (Undiluted) | Kindly gifted by Dr. Owaribe (Nagoya University, Nagoya, Japan) | N/A |
| Rat anti-cingulin mAb, IF (Undiluted) | Kindly gifted by Dr. Furuse (NIPS, Okazaki, Japan) | N/A |
| Mouse anti-ZO1mAb, IF (Undiluted), IEM (Undiluted) | Itoh *et al* (1991) | N/A |

**Reagents and Tools table**  (continued)

| Reagent/Resource | Reference or Source | Identifier or Catalog Number |
|---|---|---|
| Rat anti-ZO1 mAb, IF (Undiluted) | Kitajiri *et al* (2004) | N/A |
| Rat anti-occludin mAb, IEM (Undiluted) | Saitou *et al* (1997) | N/A |
| Rabbit anti-LUZP1(N) pAb, IF (1:200), WB (1:500), IEM (1:100) | This study | N/A |
| Rabbit anti-LUZP1(M) pAb, IF (1:200), WB (1:500) | This study | N/A |
| Rabbit anti-LUZP1(C) pAb, IF (1:200), WB (1:500) | This study | N/A |
| Rat anti-LUZP1 (N) pAb, IF (1:200), WB (1:500) | This study | N/A |
| Rat anti-LUZP1 (M) pAb, IF (1:200), WB (1:500) | This study | N/A |
| Rat anti-LUZP1 (C) pAb, IF (1:200), WB (1:500) | This study | N/A |
| Donkey anti-Rabbit IgG (H + L), Alexa Fluor 488, IF (1:1,000) | Molecular Probes | Cat#A-21206; RRID: AB_2535792 |
| Donkey anti-Rabbit IgG (H + L), Alexa Fluor 568, IF (1:1,000) | Molecular Probes | Cat#A10042; RRID: AB_2534017 |
| Donkey anti-Rabbit IgG (H + L), Alexa Fluor 647, IF (1:1,000) | Molecular Probes | Cat#A-31573; RRID: AB_2536183 |
| Donkey anti-Rat IgG (H + L), Alexa Fluor 488, IF (1:1,000) | Molecular Probes | Cat#A-21208; RRID: AB_141709 |
| Donkey anti-Rat IgG (H + L), Alexa Fluor 594, IF (1:1,000) | Molecular Probes | Cat#A-21209; RRID: AB_2535795 |
| Donkey anti-Mouse IgG (H + L), Alexa Fluor 488, IF (1:1,000) | Molecular Probes | Cat#A-21202; RRID: AB_141607 |
| Donkey anti-Mouse IgG (H + L), Alexa Fluor 568, IF (1:1,000) | Molecular Probes | Cat#A10037; RRID: AB_2534013 |
| Donkey anti-Mouse IgG (H + L), Alexa Fluor 647, IF (1:1,000) | Molecular Probes | Cat#A-31571; RRID: AB_162542 |
| Donkey anti-Goat IgG (H + L), Alexa Fluor 488, IF (1:1,000) | Molecular Probes | Cat#A-11055; RRID: AB_2534102 |
| Donkey anti-Goat IgG (H + L), Alexa Fluor 568, IF (1:1,000) | Molecular Probes | Cat#A-11057; RRID: AB_142581 |
| Goat anti-Rabbit IgG (H + L), HRP conjugate, WB (1:2,000) | GE Healthcare | Cat#NA934; RRID: AB_772206 |
| Goat anti-Rat IgG (H + L), HRP conjugate, WB (1:2,000) | GE Healthcare | Cat#NA935; RRID: AB_772207 |
| Goat anti-Mouse IgG (H + L), HRP conjugate, WB (1:2,000) | GE Healthcare | Cat#NA931, RRID: AB_772210 |
| Goat anti-Mouse IgG Antibody (H + L), Biotinylated | Vector Laboratories | Cat#BA-9200; RRID: AB_2336171 |
| Nanogold-Fab Goat anti-Rabbit IgG, IEM (1:10) | Nanoprobes | Cat#2004; RRID: AB_2631182 |
| Alexa Fluor® 546 - FluoroNanogold™ Fab' Goat anti-Mouse IgG (H + L), IEM (1:10) | Nanoprobes | Cat#7402; RRID: AB_2631183 |
| Mouse IgG-UNLB antibody | Southern Biotech | Cat#0107-01; RRID: AB_2732898 |
| **Oligonucleotides and other sequence-based reagents** | | |
| PCR primers | This study | Appendix Table S1 |
| **Chemicals, enzymes and other reagents** | | |
| Rhodamine Phalloidin | Molecular Probes | Cat#R415; RRID: AB_2572408 |
| Y27632 | Wako | Cat#257-00511 |
| Calyculin A | Cell Signaling Technology | Cat#9902 |
| Nocodazole | Sigma-Aldrich | Cat#M1404 |

**Reagents and Tools table**  (continued)

| Reagent/Resource | Reference or Source | Identifier or Catalog Number |
|---|---|---|
| Fetal bovine serum | Nichirei Biosciences | Cat#171012 |
| Lipofectamine 2000 | Life Technologies | Cat#11668027 |
| PEI MAX | Polysciences | Cat#24765-1 |
| 293fectin transfection reagent | Invitrogen | Cat#12347019 |
| Cellfectin II | Invitrogen | Cat#10362100 |
| G 418 Disulfate Aqueous Solution | Nacalai tesque | Cat#09380-44 |
| Puromycin | Sigma-Aldrich | Cat#P8833 |
| Taxol | Sigma-Aldrich | Cat#T7191 |
| ATP | Sigma-Aldrich | Cat#A2383 |
| GTP | Sigma-Aldrich | Cat#G8877 |
| Protease inhibitor cocktail | Nacalai tesque | Cat#03969 |
| Poly/Bed 812 | Polysciences | Cat#08791-500 |
| Tubulin | Cytoskeleton Inc. | Cat#T240 |
| GST-ROCK1-catalytic domain | Carna biosciences | Cat#01-109 |
| Recombinant human PAK1 protein | BPS Bioscience | Cat#40072 |
| ANTI-FLAG M2 Affinity Gel purified immunoglobulin | Sigma-Aldrich | Cat#A2220 |
| 3X FLAG Peptide | Sigma-Aldrich | Cat#F4799 |
| FLAG-LUZP1 | This study | N/A |
| Protein A Sepharose CL-4B | GE Healthcare | Cat#A11120 |
| Glutathione Sepharose 4B Fast Flow | GE Healthcare | Cat#17-5132 |
| GST-LUZP1 | This study | N/A |
| GST-MLC-WT | This study | N/A |
| GST-MLC-DD | This study | N/A |
| GST-MLC-AA | This study | N/A |
| GST-Merlin | This study | N/A |
| GST-PP1c | This study | N/A |
| T4 DNA Ligase | Promega | Cat#M1801 |
| Immobilon Western Chemiluminescent HRP Substrate | Millipore | Cat#WBKLS |
| Block Ace | DS Pharma Biomedical | Cat#UK-B80 |
| HQ Silver Enhancement Kit | Nanoprobes | Cat#2012 |
| Faramount Aqueous Mounting Medium | DAKO | Cat#S3025 |
| **Software** | | |
| Fiji | NIH | SCR_002285 |
| Metamorph Imaging Software | Molecular Devices | SCR_002368 |
| R | R Development Core Team | http://www.r-project.org |
| **Other** | | |
| Q-Sepharose Fast Flow | GE Healthcare | Cat#17051001 |
| SP Sepharose Fast Flow | GE Healthcare | Cat#17072901 |
| Dulbecco's modified Eagle's medium (DMEM) | Nissui Pharmaceuticals | Cat#05919 |
| Sf-900 III serum-free medium | GIBCO | Cat#12658027 |

## Methods and Protocols

### Ethics statement
Animal experiments were performed in accordance with protocols approved by the Animal Experiment Committee of Osaka University and Kyoto University. Recombinant DNA experiments were carried out in accordance with protocols approved by Osaka University.

### Identification of LUZP1 by the membrane overlay assay
The AJC-enriched bile canaliculus fraction was prepared from 2-day-old chick liver (Tsukita & Tsukita, 1989; Yamazaki *et al*, 2008; Yano *et al*, 2013), suspended in a hypotonic solution (1 mM $NaHCO_3$, 4 µg/ml leupeptin, pH 7.5), and then ultra-centrifuged at 100,000 *g* at 4°C for 30 min (Beckman Coulter type 45 Ti rotor). The pellet was then suspended in buffer A (10 mM HEPES [pH 7.5], 1 mM EGTA, 6 M urea, 4 µg/ml leupeptin, 10 mM APMSF; "1: Ho") and was ultra-centrifuged at 100,000 *g* at 4°C for 60 min (Beckman Coulter type TLA100.3 rotor; Supernatant, "2: Sup"; Pellet, "3: Pel"). The supernatant ("2: Sup") was applied to an SP-Sepharose column (#17072901; GE Healthcare; flow-through fraction, "4: SP-Pass"), and fractions eluted between with 100 mM and 150 mM NaCl were collected in a buffer A (50 mM HEPES [pH 7.5], 1 mM EGTA, 6 M urea, 2 µg/ml leupeptin, 10 mM APMSF) to obtain "SP-E3(1)", "5: SP-E3(2)", and "SP-E3(3)". The "5: SP-E3(2)" fraction was applied to a Q-Sepharose column (#17051001; GE Healthcare; flow-through fraction, "6: Q-Pass"), and fractions were eluted with 50 mM NaCl in buffer A to obtain "7: Q-E3(1)", "8: Q-E3(2)", and "9: Q-E3(3)". The fractions were subjected to SDS–PAGE and transferred to the polyvinylidene difluoride (PVDF) membranes. Membranes were blocked with 1% bovine serum albumin (BSA) in PME buffer (80 mM PIPES, 1 mM $MgCl_2$, 1 mM EGTA; pH 6.9) at room temperature (RT) for 60 min and then incubated with microtubule (MT) solution in the presence of 5% skim milk. For MT solution, 1 mg/ml tubulin purified from the porcine brain was polymerized at 37°C in a polymerization buffer (3 mM $MgCl_2$, 1 mM EGTA, 1 mM GTP, 10% DMSO, 80 mM PIPES; pH 6.8) for 60 min and then diluted 22-fold with 20 µM Taxol (#T7191; Sigma-Aldrich) containing PME buffer. After washing with PME buffer at 37°C for 5 min three times, the membrane was fixed with 10% trichloro-acetic acid in Milli-Q water (Millipore) at 4°C for 10 min and washed with TBS (10 mM Tris–HCl [pH 7.5], 150 mM NaCl) at RT for 5 min three times. Proteins binding to polymerized MTs were detected by mouse anti-α-tubulin antibody (#T9026; Sigma-Aldrich), followed by biotin-conjugated anti-mouse IgG antibody (#BA9200; Vector Laboratories) and alkaline phosphatase-conjugated streptavidin using NBT/BCIP visualization. The corresponding single band of about 150 kDa molecular weight protein in the silver-stained polyacrylamide gel was cut out, and its amino acid sequence was determined by Edman degradation (APRO Science).

### Cell culture, transfection, chemical treatments, and sample preparation for immunoblotting
Mouse mammary gland epithelial Eph4 cells (Reichmann *et al*, 1989), MTD-1A cells (epithelial cells derived from malignant neoplasms of the mouse mammary gland; Fig 2I) (Hirano *et al*, 1987), Human Embryonic Kidney cells 293 (HEK-293) cells (Graham *et al*, 1977), and CSG120/7 cells (epithelial cells derived from malignant neoplasms of the mouse submandibular gland; Fig 2J)

(Knowles & Franks, 1977) were cultured in Dulbecco's modified Eagle's medium (DMEM; #05919; Nissui Pharmaceuticals) supplemented with 10% fetal bovine serum (FBS; #171012; Nichirei Biosciences) at 37°C and 5% $CO_2$. Insect Sf9 cells were cultured in Sf-900 III serum-free medium (#12658027; GIBCO) supplemented with 10% FBS at 27°C. Transfection was performed using Lipofectamine 2000 (#11668019; Invitrogen), PEI MAX (#24765-1; Polysciences), 293fectin transfection reagent (#12347019; Invitrogen), or Cellfectin II (#10362100; Invitrogen) as appropriate following the manufacturer's instructions. For establishing stable transfectants, the transfected Eph4 cells were selected by incubation in medium containing 500 µg/ml G418 (#9380-44; Nacalai Tesque) and cell clones derived from single cells were picked up. For chemical treatment, the cells were incubated in DMEM containing 100 µM Y27632 (#257-00511; Wako) for 30 or 60 min, 100 nM calyculin A (#9902; Cell Signaling Technology) for 30 min, or 2 µM nocodazole (#M1404; Sigma-Aldrich) for 30 min. For immunoblotting of cultured cells, confluent cells on each 6-cm dish were washed three times with HBS (10 mM HEPES, 150 mM NaCl; pH 7.5) and scraped off plate with 500 µl SDS sample buffer (50 mM Tris–HCl [pH 6.8], 2% SDS, 10% glycerol, 2% β-mercaptoethanol, and 0.02% bromophenol blue). The samples were sonicated (Sonifier 250; Branson), boiled at 98°C for 10 min, and centrifuged at 20,400 *g* at 20°C for 10 min. The supernatants were collected, and protein concentrations were determined using BSA as a standard. For immunoblotting of mouse tissues, each tissue was carefully collected out from 15-week-old C57BL/6J mice, immediately frozen in liquid nitrogen, and homogenized with SK-100 (Tokken). Then, SDS sample buffer without 10% glycerol (50 mM Tris–HCl [pH 6.8], 2% SDS, 2% β-mercaptoethanol, and 0.02% bromophenol blue) was added. The samples were sonicated (Sonifier 250; Branson) and centrifuged at 20,400 *g* at 4°C for 30 min. Glycerol was added to the supernatant (glycerol final concentration: 10%), and protein concentrations were determined using BSA as a standard.

### SDS–PAGE and immunoblotting
Equal amounts of proteins were separated via 7.5%, 12.5%, 4–7.5%, or 10–20% SDS–PAGE gel and transferred to PVDF membranes. The PVDF membranes were blocked with 5% skim milk or 5% BSA for 30 min, probed with primary antibodies at RT for 60–120 min, and then incubated with HRP-conjugated secondary antibodies at RT for 30–60 min. Immunoblots were developed using an enhanced chemiluminescence kit (#WBKLS0500; Millipore). Densitometric quantification of the SDS–PAGE bands or immunoblotted bands was performed using the "Gel Analyzer" module in ImageJ (freely available at https://imagej.nih.gov/ij/index.html).

### Generation of LUZP1 knockout (LUZP1 KO) cells, ZO-1/-2 DKO cells, and E-cadherin KO cells with CRISPR/Cas9 system and generation of Venus-LUZP1-expressing LUZP1 knockout (REV) cells
To generate LUZP1 KO, ZO-1/-2 DKO, and E-cadherin KO Eph4 cells, we used the CRISPR/Cas9 system with the pX330 vector (#42330; Addgene) to knockout mouse LUZP1, ZO-1, and E-cadherin genes. Targeting sequences of guide RNAs of LUZP1 (5′-GGCAGAACTCACTAACTACA-3′, 5′-GGATGAGCTCCTGGACCTCC-3′, or 5′-GCTCCTGGACCTCCAGGACA-3′), ZO-2 (5′-GCAGCGCGGTCCAGGCATG-3′), and E-cadherin (5′- GGTCTACACCTTCCC

GGTGC-3′) were annealed and cloned into the BbsI site of the pX330 vector using T4 DNA ligase (#M1801; Promega; Please refer to Appendix Table S1 and S2 for the further information on oligonucleotides used in this study). Eph4 and ZO-1 KO Eph4 cells (Umeda *et al*, 2004) were transfected with LUZP1 KO, ZO-2 KO, and E-cadherin KO pX330 plasmids, respectively, using Lipofectamine 2000 according to manufacturer's instructions. LUZP1 KO and ZO-1/-2 DKO cells were sorted and isolated by limiting dilution. Single cell-derived KO and DKO lines were confirmed by genomic sequence analyses. For the generation of REV cells, at first, mouse LUZP1 full-length cDNA was inserted into pCAGGS-Venus vector, which was constructed by the insertion of a Venus-tag into the neomycin-resistant pCAGGS vector (kindly gifted by Hitoshi Niwa), to obtain Venus-LUZP1 plasmid. Then, Venus-LUZP1 plasmids were transfected into LUZP1 KO Eph4 cells using Lipofectamine 2000 following the manufacturer's instructions. The transfected Eph4 cells were selected by incubation in medium containing 500 μg/ml G418, and cell clones derived from single cells were picked up to establish stable REV cells.

### Purification of Flag-LUZP1 using HEK-293 cells

Mouse LUZP1 full-length cDNA was inserted into the pCAGGS-Flag vector, which was constructed by the insertion of a Flag-tag into the pCAGGS vector (kindly gifted by Hitoshi Niwa), to obtain Flag-LUZP1 plasmid. HEK-293 cells on each 10-cm dish were transfected with 3 μg Flag-LUZP1 plasmid using PEI MAX (Polysciences) following the manufacturer's instructions. The cells in each 10-cm dish were washed three times with ice-cold HBS and then scraped with 300 μl RIPA buffer (150 mM NaCl, 0.1% SDS, 0.5% deoxycholic acid, 1% Nonidet P-40, 50 mM Tris–HCl [pH 8.0], and protease inhibitor cocktail [#03969; Nacalai Tesque]). The cell lysate obtained from twenty 10 cm-diameter dishes was centrifuged at 20,400 *g* at 4°C for 30 min and incubated with 20 μl anti-Flag M2 affinity gel beads (#A2220; Sigma-Aldrich) at 4°C for 120 min. After incubation, the beads were washed five times with ice-cold HBS and eluted with 3X-FLAG peptide (#F4799; Sigma-Aldrich) in ice-cold HBS following the manufacturer's instructions. Protein solutions were dialyzed in cellulose dialysis membrane, which is soaked in ice-cold HBS to remove the 3X-FLAG peptide at 4°C for overnight.

### Microtubule co-sedimentation assay

Tubulin protein (#T240; Cytoskeleton Inc.) was polymerized in 1 mM GTP and 1 mM Taxol-containing PME buffer at 37°C for 60 min (tubulin final concentration: 12 μM). Microtubule co-sedimentation assay was carried out in a reaction volume of 20 μl. For full-length LUZP1, 10 μl of 7.7 nM Flag-LUZP1 in HBS was mixed with 10 μl of each diluted tubulin solution (final tubulin concentration in 20 μl: 0, 0.375, 0.75, 1.5, 3, or 6 μM) in 1 mM GTP- and 1 mM- Taxol-containing PME buffer and incubated at RT for 60 min. For LUZP1 regions, 10 μl of 20 nM GST-LUZP-N-terminal, middle, and C-terminal regions in HBS was mixed with 10 μl of tubulin solutions (final tubulin concentration in 20 μl: 10 μM) in 1 mM GTP- and 1 mM Taxol-containing PME buffer and incubated at RT for 60 min. After incubation, samples were then ultra-centrifuged at 434,513 *g* at 25°C for 20 min (Beckman Coulter type TLA100 rotor). Pellets were resuspended in a 1:1 mixture of 1 mM GTP- and 1 mM Taxol-containing PME buffer and HBS. After adding SDS sample buffer, both supernatants and pellets were boiled at 98°C for 10 min and subjected to SDS–PAGE. The proteins were evaluated by immunoblotting. Densitometric quantification of immunoblot signal was performed using the "Gel Analyzer" module in ImageJ. The MT-bound-LUZP1 fraction (*y*) was plotted against tubulin concentration (*x*), and the dissociation constant ($K_d$) value of each experiment was determined from the best fitted curve to the Michaelis–Menten equation.

### Immunoprecipitation

HEK-293 cells on each 6-cm dish were transiently co-transfected with 1.5 μg of HA- or Venus-tagged WT or mutant LUZP1 plasmid and 1.5 μg of another HA-, GFP-, or Venus-tagged protein plasmid, as appropriate. The cells on each 6-cm dish were washed three times with ice-cold HBS and then scraped with 300 μl RIPA buffer or Hypo buffer (1mM NaHCO₃; pH 7.5). Hypo buffer-treated cells were incubated on ice for 10 min and sonicated. The 150 mM NaCl adjusted-cell lysate, clarified by centrifugation at 20,400 *g* at 4°C for 30 min, was incubated with 20 μl of protein A-Sepharose bead slurry (#17-0780-01; GE Healthcare) conjugated in advance with mouse anti-GFP antibodies (#A11120; Life Technologies) or mouse anti-HA antibodies (#MMS-101R; Covance), or total mouse IgG (#0107-01; Southern Biotech) as a control, at RT for 120 min. After five washes with RIPA buffer or TBS, the immunoprecipitation beads were dissolved in 40 μl SDS sample buffer, followed by separation via SDS–PAGE. The immunoprecipitated proteins were evaluated by immunoblotting. Densitometric quantification of immunoblot signals was performed using the "Gel Analyzer" module in ImageJ.

### Immunofluorescence microscopy

Cells plated on glass coverslips were fixed in cold methanol at −20°C for 10 min or fixed in 1% formaldehyde in HBS at RT for 8 min. The fixed cells were treated with 0.25% Triton X-100 in HBS at RT for 5 min and washed three times with HBS. After soaking in HBS containing 1% bovine serum albumin (BSA) at RT for 10 min, the samples were treated with primary antibodies at RT for 60–180 min, followed by washing three times with HBS and incubating with secondary antibodies at RT for 60 min (In some experiments, rhodamine-phalloidin [#R415; Molecular Probes] was added to detect F-actin). The samples were washed three times with HBS, shortly soaked in Milli-Q water (Millipore), and mounted with Fara-mount Mounting Medium (#S3025; DAKO). Immunofluorescent micrographs were acquired using a fluorescence microscope (BX51 or BX53; Olympus), a fluorescence microscope with a disk scanning unit system (BX53-DSU; Olympus), or a Spinning Disk Confocal Super Resolution Microscope (SD-OSR; Olympus).

### Analysis of immunofluorescent micrographs

For outlining individual cells, marking regions of interests (ROIs) of each cell, and making individual cell-outlined binary images, we used ZO-1 immunostained micrographs and a plugin in ImageJ (Automated Multicellular Tissue Analysis) developed by the Advanced Digital Microscopy Core Facility at the Institute for Research in Biomedicine (Barcelona, Spain; freely available at http://adm.irbbarcelona.org). To calculate apical or basal areas of the cell, we used the top and bottom images of cell sheets with the *z*-series of several planes, 0.25-μm apart. To calculate the mean fluorescent intensity within CRs of each cell, we first

marked ROIs of CRs using the ROIs of each cell, individual cell-outlined binary images, and "dilate" function in ImageJ. Then, we obtained the mean intensity of a protein of interest in each ROI to quantify a mean fluorescent intensity within CRs of individual cells. For calculation of the degree of co-localization between two specific proteins, we calculated Pearson's correlation coefficients within each cell using ROIs of each cell and the plugin called "JACoP" (another co-localization plugin) in ImageJ. For plotting signal intensity along the line, we used the "plot profile" function in ImageJ.

### Immunoelectron microscopy

Cells were fixed in 1% formaldehyde in HBS at RT for 8 min and permeabilized by 5% saponin in HBS at RT for 10 min. After blocking in a blocking buffer containing 5% saponin in Block Ace (#UK-B80; DS Pharma Biomedical) at RT for 5 min, the cells were incubated with primary antibodies in blocking buffer at 37°C for 120 min, followed by incubation with secondary antibodies diluted in blocking buffer at 37°C for 120 min. The cells were then fixed in a solution containing 2% paraformaldehyde, 2.5% glutaraldehyde, 0.5% tannic acid, and 0.1 M HEPES buffer at RT for 60 min and washed with 0.1 M HEPES buffer (pH 7.5) and then with 50 mM HEPES buffer (pH 5.8). The cells were then mounted in the reagent from the HQ Silver Enhancement Kit (#2012; Nanoprobes). Cell lipids were fixed with 1% $OsO_4$ in 0.1 M HEPES buffer (pH 7.5) on ice for 120 min. The samples were then dehydrated and embedded in Poly/Bed 812 (#08791-500; Polysciences). Ultra-thin sections were imaged using a transmission electron microscope (JEM-1400 plus; JEOL).

### In vitro binding assay between LUZP1 and myosin light chain (MLC)

Wild-type MLC (WT-MLC), di-dephosphomimetic MLC (AA-MLC), and di-phosphomimetic MLC (DD-MLC) were constructed as described previously (Iwasaki *et al*, 2001). Using pGEX vector system, GST-tagged WT-MLC, AA-MLC, and DD-MLC were produced in BL21 *Escherichia coli* at 16°C for overnight following the manufacturer's instructions. Transformed BL21 *E. coli* were sonicated (Sonifier 250; Branson), and lysates were centrifuged at 21,500 *g* at 4°C for 30 min. The supernatants were then conjugated with 500 µl Glutathione Sepharose 4B (#17-5132; GE Healthcare) at 4°C for 120 min. Beads were washed with HBS five times and eluted with 10 mM glutathione in 50 mM Tris buffer (pH 8.0). Protein solutions were dialyzed in cellulose dialysis membrane soaked in HBS to remove glutathione at 4°C for overnight.

The *in vitro* binding assay was carried out in a reaction volume of 8 µl. 3 µl of 11 ng/µl Flag-LUZP1-conjugated anti-Flag M2 affinity gel beads (#A2220; Sigma-Aldrich) in HBS was mixed with 5 µl of GST-tagged WT-MLC, AA-MLC, or DD-MLC in HBS (final concentration in 8 µl: 0, 0.5, 1.0, 2.0, 4.0, and 8.0 µM) and incubated at RT for 60 min. Beads were washed twice in washing buffer (300 mM NaCl, 50 mM Tris–HCl [pH 7.5], 1 mM EDTA, and 1 mM DTT) and finally dissolved in 20 µl SDS sample buffer. Samples were subjected to SDS–PAGE and the proteins were evaluated by immunoblotting. Densitometric quantification of immunoblot signals was performed using the "Gel Analyzer" module in ImageJ. In each assay, the relative ratio of MLC/LUZP1 signal intensity ($y$) was plotted against MLC concentrations ($x$), and the $K_d$ value of

each experiment was determined from the best fitted curve of the Michaelis–Menten equation.

### Purification of recombinant mouse LUZP1 using the Bacmid system

Mouse LUZP1 full-length cDNA was inserted into a pFastBac-based plasmid that was constructed by inserting a GST coding sequence and a multi-cloning site from the pGEX6P2 vector (#28-9546-50; GE Healthcare) into pFastBac1 vector (#10712024; Invitrogen). This plasmid was transformed into DH10Bac *E. coli* (#10361012; Invitrogen) to obtain a bacmid. Insect Sf9 cells were then transfected with the bacmid using Cellfectin II (#10362100; Invitrogen). After 3 days of transfection, the culture medium containing baculovirus was collected and added to other Sf9 cells in 6-well dishes for infection. After 3 days of infection, Sf9 cells were lysed with RIPA buffer (100 µl/well). The cell lysate, clarified by centrifugation at 20,400 *g* at 4°C for 30 min, was incubated with 500 µl glutathione Sepharose 4B (#17-5132; GE Healthcare) at 4°C for 120 min. After five washes with RIPA buffer, beads were eluted with 10 mM glutathione in 50 mM Tris buffer (pH 8.0). Protein solutions were dialyzed in cellulose dialysis membrane soaked in ice-cold HBS to remove glutathione at 4°C overnight.

### Phosphorylation assay

Phosphorylation assays were carried out in a 12.5 µl (MLC phosphorylation assay) or 15 µl (Merlin phosphorylation assay) reaction mixture (130 mM KCl, 20 mM NaCl, 1 mM $MgCl_2$, 1 mM ATP, 10 mM HEPES, 1 mM GTP, 1 mM Taxol, pH 7.5). For the MLC phosphorylation assay, 2 µl of 12.5 ng/µl GST-MLC purified from BL21 *E. coli* was mixed with 2 µl of 2 ng/µl GST-ROCK1-catalytic domain (#01-109; Carna biosciences), 1 µl of 1 µg/µl GST-protein phosphatase 1c (GST-PP1c) purified from BL21 *E. coli*, and 0, 0.5, 2.0, or 5.0 µl of 1 µg/µl GST-LUZP1 purified using the bacmid system, with or without 1 µl of 1 µg/µl tubulin (#T240; Cytoskeleton Inc.), which was polymerized in advance in PME buffer with 1 mM GTP and 1 mM Taxol at 37°C for 60 min. For the Merlin phosphorylation assay, 2 µl of 50 ng/µl GST-Merlin purified from BL21 *E. coli* was mixed with 2 µl of 1 pg/µl recombinant human PAK1 protein (#40072; BPS Bioscience), 1 µl of 1 µg/µl GST-protein phosphatase 1c (GST-PP1c) purified from BL21 *E. coli*, and 0, 0.5, 2.0, or 5.0 µl of 1 µg/µl GST-LUZP1 purified using the bacmid system, with or without 1 µl of 1 µg/µl tubulin, which was polymerized in advance in PME buffer with 1 mM GTP and 1 mM Taxol at 37°C for 60 min.

The mixtures were incubated at 37°C for 60 min. After incubation, the samples were subjected to SDS–PAGE in 5–20% polyacrylamide gels and transferred to the PVDF membranes. The membranes were then blocked with 5% skim milk, probed with primary antibodies at RT for 60–120 min, and then incubated with HRP-conjugated secondary antibodies for 30–60 min. Immunoreactive signals were developed using enhanced chemiluminescence kits (#WBKLS; Millipore). Densitometric quantification of the SDS–PAGE or immunoblot signals was performed using the "Gel Analyzer" module in ImageJ.

### Quantification and statistical analysis

All of the experiments were repeated multiple times as indicated in the figure legends. Statistical analysis was performed with at least on three biological replicates under similar conditions using the

statistical software R, version 3.4.0 (R Development Core Team, 2017, freely available at https://www.R-project.org). We firstly examined whether the data were normally distributed using the Shapiro–wilk test. Then, as appropriate, we used an unpaired *t*-test or Mann–Whitney *U* test for comparisons between two groups, one-way analysis of variance (ANOVA) followed by Tukey–Kramer test or the Kruskal–Wallis test followed by the Steel test or the Steel–Dwass test for comparisons between multiple groups, and Pearson's correlation coefficient for correlation analysis. Data are presented as the mean ± SD. *P*-values < 0.05 were considered statistically significant.

## Data availability

Statistical source data used were deposited in OSF (Open Science Framework). View-only Links to our data is https://osf.io/pv5me/?view_only = 994cade5cbe34997ad9fc9fe5b41fdeb.

**Expanded View** for this article is available online.

## Acknowledgements
We would like to thank Dr. Masami Uji, Ms. Yuki Sugiyama, and Ms. Fumiko Takenaga for an excellent technical assistance. We are grateful to our laboratory members, Dr. Naruki Sato, and Dr. Tadashi Uemura for discussions and encouragement. We would also like to thank Dr. Marius Sudol for careful proofreading. We also thank Lifematics Inc. for graphical support. We appreciate support of Dr. Hiroko Okinaga, Dr. Koji Aoyama, and Dr. Yoshihito Okinaga of Teikyo University. This work was supported by Core Research for Evolutionary Science and Technology CREST of the Japan Science and Technology Agency (JST) (JPMJCR13W4 to Sachiko Tsukita), and by Grant-in-Aid for Specially Promoted Research (JP19H05468, to Sachiko Tsukita), Grant-in-Aid for Early-Career Scientists (JP18K14696, to Tomoki Yano), and Grant-in-Aid for Scientific Research (B) (JP16H05121, to Atsushi Tamura) from Japan Society for the Promotion of Science (JSPS) of Japan.

## Author contributions
Conceptualization: TY, KT, HKan, AT, and ST; Work design: TY, KT, and ST; Data acquisition: TY, KT, HKan, SN, TMi, HKas, TMa, YG, AK, and KA; Data analysis: TY, KT, TMi, AT, and ST; Data interpretation: TY, KT, SN, TMi, KA, AT, and ST; Manuscript drafting: TY, KT, HT, RT, AT, and ST.

## Conflict of interest
The authors declare that they have no conflict of interest.

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
