## [Review Process File · The EMBO Journal]

A microtubule-LUZP1 association around tight junction promotes epithelial cell apical constriction

Tomoki Yano, Kazuto Tsukita, Hatsuho Kanoh, Shogo Nakayama, Hiroka Kashihara, Tomoaki Mizuno, Hiroo Tanaka, Takeshi Matsui, Yuhei Goto, Akira Komatsubara, Kazuhiro Aoki, Ryosuke Takahashi, Atsushi Tamura, and Sachiko Tsukita

DOI: doi.org/10.15252/embj.2020104712

Corresponding authors: Sachiko Tsukita (tsukiweb@biosci.med.osaka-u.ac.jp) , Atsushi Tamura (atamura@biosci.med.osaka-u.ac.jp)

Review Timeline:

Submission Date:	15th Feb 20
Editorial Decision:	16th Mar 20
Revision Received:	9th Aug 20
Editorial Decision:	17th Sep 20
Revision Received:	2nd Oct 20
Accepted:	14th Oct 20

Editor: Ieva Gailite

Transaction Report:

Thank you for submitting your manuscript for consideration by the EMBO Journal. We have now received three referee reports on your manuscript, which are included below for your information.

As you will see from the comments, all reviewers appreciate the work and the topic. However, they also raise a number of partially overlapping concerns that need to be addressed before they can support publication here. From my side, I judge the referee comments to be generally reasonable. Therefore, based on the overall interest expressed in the reports, I would like to invite you to submit a revised version of your manuscript in which you address the comments of all three referees. I should add that it is The EMBO Journal policy to allow only a single major round of revision and that it is therefore important to resolve the main concerns at this stage.

We generally allow three months as standard revision time. However, we are aware that many laboratories cannot function at full efficiency during the current COVID-19/SARS-CoV-2 pandemic and we have therefore extended our 'scooping protection policy' to cover the period required for a full revision to address the essential experimental issues. This means that competing manuscripts published during revision period will not negatively impact on our assessment of the conceptual advance presented by your study. Please contact me if you need additional revision time, which can be easily arranged, and also if you see a paper with related content published elsewhere.

When preparing your letter of response to the referees' comments, please bear in mind that this will form part of the Review Process File, and will therefore be available online to the community. For more details on our Transparent Editorial Process, please visit our website:

<https://www.embopress.org/page/journal/14602075/authorguide#transparentprocess>

Please feel free to contact me if you have any further questions regarding the revision. Thank you for the opportunity to consider your work for publication. I look forward to receiving your revised manuscript.

Referee #1:

This study focuses on mechanisms of apical constriction, a common cell behavior that involves actomyosin-based circumferential rings (CRs). Yano et al. identified LUZP1, a microtubule (MT)-binding protein that localizes at the CRs near tight junctions (TJs). LUZP1 promotes apical constriction in epithelial Eph4 cells and is required for diphosphorylation of the regulatory myosin light chain (ppMLC) at the CRs. LUZP1 apparently acts by inhibiting myosin phosphatase PP1c. Other experiments suggest that LUZP1 activity is facilitated by MTs.

In principle, the observations of LUZP1-mediated regulation of MLC diphosphorylation and apical constriction are novel and significant. The reduction of ppMLC in LUZP1 KO-Eph4 cells is striking and, in general, the experiments are of high quality. However, the model of LUZP1-regulated apical constriction that the authors are pursuing and the role of microtubules in LUZP1 regulation are not made clear. Is the recruitment of LUZP1 to TJ necessary for apical constriction to occur? The link to MT could be strengthened by showing the association of LUZP1 and MTs at the TJ and addressing its physiological role. Finally, the reader would like to know whether the proposed function of LUZP1 is conserved in other epithelial cells.

Other comments:

1. Does LUZP1 function the same way in other cell lines and/or normal epithelial tissues?
2. The association of LUZP1 with MT is not clear in Fig.1H. Its colocalization with MTs at the AJCs should be shown by immunostaining.
3. Since Y27632 treatment reduced LUZP1 localization at CRs (Fig.2G), the authors propose a positive feedback loop between ppMLC and LUZP1 localization at CRs. The authors have to consider alternative possibilities and explain what initiates MLC di-phosphorylation. What recruits LUZP1 to CRs? The authors show that LUZP1 interacts with ZO1, but it is unclear what role ZO1 plays in LUZP1 recruitment to TJs or CRs.
4. The authors should better explain why Merlin is a relevant substrate to assess PP1c activity. Fig.5F: PP1c does not completely suppress Pak1-mediated phosphorylation of Merlin. More direct measurement of PP1c activity is warranted.
5. Fig.3E, Fig.5EF: In vitro assay of ppMLC needs improvement, because GST-MLC bands are very weak. GST-MLC can't be 20 kDa, is this a typo? The error bars are very large in Fig.5EF. Additional controls, such as LUZP1 alone and, possibly Calyculin A treatment, are needed.
6. Figs.6AB. Apical constriction needs to better supporting images. Does nocodazole reduce ppMLC specifically at CRs or overall levels?

Referee #2:

Tsukita and colleagues have previously shown a structural and functional link between tight junctions and microtubules. Here, they extend this work uncovering a new mechanistic component, LUZP1, and extend the functional significance to apical constriction, a process contributing to epithelial cell morphogenesis in columnar epithelia and neural tube closure. While LUZP1 was known to be involved in neural tube closure, there was no information of the underlying cellular and molecular mechanisms.

This is an interesting and largely convincing study that should be of interest to a wide readership. The authors use an impressive set of complementary assays to support their conclusions that appear to be performed to a high standard. Occasionally, the paper requires some modifications to improve clarity and minor experiments to support the conclusions drawn.

1) All components tested to interact with LUZP1 bind to the same fragment of the protein, the C-terminal domain. This includes ZO-1, MLC as well as tubulin/microtubules. It is surprising that junctional LUZP1 should bind to all of these proteins with the same domain. However, inhibition of ROCK leads to junctional dissociation of LUZP1, despite its ZO-1 binding site, and does not lead to any obvious association with microtubules (Fig. 2G). Hence, one wonders which of these interactions actually occur under physiological conditions (i.e., not transfected and vastly overexpressing HEK293 cells or in a test tube). Do these proteins all co-immunoprecipitate from wild type Eph4 cells? Is binding of MLC, ZO-1 and tubulin to LUZP1 competitive? How do the authors envision how these interactions are coordinated within the junction?

2) Fig. 1C: According to the methods, the co-IP was performed with cold cell extract; hence, the interaction seen is likely to be with α -tubulin, not microtubules as they depolymerise in the cold. As binding sites for free tubulin and microtubules may be distinct, the co-sedimentation assay should be performed to identify the microtubule binding site within the LUZP1 fragments.

3) Page 8, line 14: What is the evidence that MLC itself is still at junctions upon treatment with Y27632? This should be tested by IF.

4) Figure 3C: The authors conclude that this experiment shows binding to ppMLC; however, there is no evidence presented that the co-immunoprecipitated MLC is phosphorylated. An immunoblot for ppMLC should be shown.

5) Figure 3H-L: How does one know that slower diffusion is not an intrinsic property of ppMLC (e.g., due to being part of active myosin) as opposed to be due to binding LUZP1? The assay should also be performed in the absence of LUZP1 to justify the conclusion.

Minor comments

6) Fig. 1B: What was the input, purified Flag-LUZP1? This should be specified to justify the conclusion that the interaction is direct.

7) Page 6: The description of figure 1F-J is confusing, appears to routinely refer to the wrong panels and the panels are not in order of the text.

8) Page 7, last paragraph: The 'epithelial' behaviour described may be observed in some epithelial cell lines but should not be declared as being the general behaviour of epithelial cells as they come in different types and shapes, and, in vivo, may behave rather differently.

9) Page 10, start of new section: The experiments show that TJ recruitment of LUZP1 is ROCK activity-dependent. Whether it is indeed ppMLC-dependent is not shown.

Referee #3:

In this manuscript the authors describe their findings on the protein LUZP1 and its involvement in apical constriction in vertebrate epithelial cells. The authors show that LUZP1 colocalises with myosin at the level of tight junctions in epithelial cells in culture, and that cells expressing LUZP1 can apically constrict whereas K.O. cell cannot. They go on to show that LUZP1 can bind to MRLC in a phosphorylation state dependent manner, and that it inhibits myosin phosphatase through direct binding. LUZP1 was originally identified by the authors at a microtubule-associated protein, and they show that interaction with microtubules is important for LUZP1 to exert its effect on myosin levels at tight junctions.

The apical constriction of epithelial cells is a key process during morphogenesis and organ formation, and thus a better understanding of the mechanisms and variations underlying this process are key to understanding development and should be of interest to a variety of readers. Furthermore, the interplay of actomyosin and microtubules is emerging as an important, albeit understudied, theme in morphogenesis, and as such the described mechanism is also of great interest.

The experiments presented appear well-executed and controlled, and the conclusions drawn seem well supported by the data.

I have a few general and specific comments, though, that I think would be worth addressing or commenting on to strengthen the paper further.

Firstly, the authors put particular emphasis on the notion that apical constriction in vertebrate cells works through a different mechanism than the one analysed in great detail in invertebrates such as *Drosophila* or *C.elegans*. In these invertebrate models apical constriction is driven by a highly dynamic pool of apical-medial actomyosin, whereas the role of junctional myosin in this process might rather be to provide a ratcheting mechanism to stabilise a shrunken apical shape.

The authors state 'In contrast, in vertebrate apical constriction, actomyosin-based circumferential rings (CRs) associated with the apical junctional complex (AJC), which includes tight junctions (TJs) and adherens junctions (AJs), play a central role.' in the introduction, but without citing any studies to support this.

I agree that the vast majority of support for dynamic apical-medial actomyosin driving apical

constriction stems from invertebrate systems, but there are indications that similar pools of myosin with similar functionality do also operate in vertebrates (see for instance Sumigraj, K. D., Terwilliger, M., & Lechler, T. (2018). Morphogenesis and Compartmentalization of the Intestinal Crypt, 1-35. <http://doi.org/10.1016/j.devcel.2018.03.024>). It took year and very careful in vivo live imaging in *Drosophila* for instance to uncover the apical-medial actomyosin networks and their functions. I would feel happier with a less dogmatic distinction between invertebrate and vertebrate apical constriction.

I am also still somewhat confused about actomyosin-driven constriction at the level of tight junctions compared to the usually described contractile actomyosin belt at the level of adherens junctions. Are these both present in the cells analysed here, are they continuous/overlapping? Are tight junction membrane proteins really also bona-fide adhesion receptors and force transducers between cells, which this work would imply? This should be commented on, please!

How do the authors envisage LUZP1 to function in apical constriction events during embryogenesis in vivo? Is the spatial and temporal control of apical constriction provided by the regulation of ppMLC and LUZP1 expressed in the same cells helps to amplify the myosin activation? Or is there control and regulation at the level of LUZP1 expression? In the neural tube for instance, is it only expressed when apical constriction is important for tube folding and closure?

The authors identify LUZP1 as a microtubule-binding protein and then show that it also binds to MLC as well as myosin phosphatase. Do the authors have any evidence about domains that mediate these binding interactions and how likely they are to all occur at the same time?

In Figure 2B the authors quantify the difference in apical constriction of cells in culture that express tagged LUZP1 in a mutant background in comparison to the KO cells: which cells were measured in the LUZP1-KO? The graph shows a ratio for these well above 1, i.e. they are apically expanded. Is this because they are the cells next to the apically constricted ones and thus are making up for their neighbours? Otherwise I find it hard to see why and how they should achieve this apically dilated shape!

The authors use a somewhat strange expression multiple times throughout the manuscript 'ppMLC mobilizes LUZP1...'. What do they mean by 'mobilizing'? I assume they mean 'recruits'?

Point-by-point responses to the Editor and Referees:

We wish to express our strong appreciation to the Editor and Referees for their insightful comments on our paper. We feel that our article has greatly improved through the reviewing process.

Response to Referee #1:**Main Comment:**

This study focuses on mechanisms of apical constriction, a common cell behavior that involves actomyosin-based circumferential rings (CRs). Yano et al. identified LUZP1, a microtubule (MT)-binding protein that localizes at the CRs near tight junctions (TJs). LUZP1 promotes apical constriction in epithelial Eph4 cells and is required for diphosphorylation of the regulatory myosin light chain (ppMLC) at the CRs. LUZP1 apparently acts by inhibiting myosin phosphatase PP1c. Other experiments suggest that LUZP1 activity is facilitated by MTs. In principle, the observations of LUZP1-mediated regulation of MLC diphosphorylation and apical constriction are novel and significant. The reduction of ppMLC in LUZP1 KO-Eph4 cells is striking and, in general, the experiments are of high quality. However, the model of LUZP1-regulated apical constriction that the authors are pursuing and the role of microtubules in LUZP1 regulation are not made clear. Is the recruitment of LUZP1 to TJ necessary for apical constriction to occur? The link to MT could be strengthened by showing the association of LUZP1 and MTs at the TJ and addressing its physiological role. Finally, the reader would like to know whether the proposed function of LUZP1 is conserved in other epithelial cells.

Response:

Thank you for your insightful comments. We tried our best to address your abovementioned concerns one by one as follows:

1) Is the recruitment of LUZP1 to TJ necessary for apical constriction to occur?

Yes, it is necessary. To confirm the importance of LUZP1 recruitment to TJs, we generated ZO-1/-2 double knockout (DKO) Eph4 cells that are known to be TJ deficient (Umeda et al, 2006; Ikenouchi et al, 2007; Otani et al, 2019). We subsequently found that LUZP1 junctional localization was disrupted in TJ-deficient ZO-1/-2 DKO cells, confirming the crucial role of TJs in the recruitment of LUZP1 to TJ-associated CRs (Fig 1I and EV2E). Co-cultures of WT and TJ-deficient ZO-1/-2 DKO cells showed that ppMLC levels within CRs were also reduced in ZO-1/-2 DKO cells (Fig 2G) and an apical area was larger in ZO-1/-2 DKO cells (Fig 2H), altogether confirming the importance of LUZP1 recruitment to TJ-associated CRs in maintaining ppMLC levels within CRs and inducing apical constriction.

2) The link to MT could be strengthened by showing the association of LUZP1 and MTs at the TJ and addressing its physiological role.

To confirm the association of LUZP1 and MTs at TJs, we re-acquired the immunofluorescence image (Fig 1F) and conduct a co-immunoprecipitation assay of wild-type (WT) Eph4 cells using the anti-LUZP1 antibody (Fig EV1I). Representative immunofluorescence images showed that LUZP1 associates with MTs at the level of ZO-1 delineated TJs (Fig 1F). Furthermore, the LUZP1-immunoprecipitated fraction of WT cells included α -tubulin, ZO-1, and MLC, confirming that LUZP1 associates with MTs and CRs at the level of TJs (Fig EV1I). Regarding the physiological importance of the association between LUZP1 and MTs, we believe that this was demonstrated by the finding that MT depolymerization reduced ppMLC levels within CRs in WT cells but not in LUZP1 KO cells (Fig 6B and C), showing that MT upregulates ppMLC within CRs via LUZP1. Altogether, these findings allow us to conclude that MT-LUZP1 association at TJs is critical for maintaining ppMLC levels within CRs.

3) Finally, the reader would like to know whether the proposed function of LUZP1 is conserved in other epithelial cells.

To examine whether the LUZP1-mediated upregulation of ppMLC levels with CRs is conserved in other epithelial cells, we knocked out LUZP1 in MTD-1A cells (epithelial cells derived from malignant neoplasms of the mouse mammary gland) and CSG120/7 cells (epithelial cells derived from mouse submandibular gland carcinoma cells). Notably, LUZP1 KO resulted in the down-regulation of ppMLC levels within CRs in both cell lines (Fig 2I and J), suggesting that the proposed function of LUZP1 is conserved in other epithelial cells.

Other comments:

1. Does LUZP1 function the same way in other cell lines and/or normal epithelial tissues?

Response:

As mentioned above in response (3) to the main comment, we generated and examined LUZP1 KO MTD-1A and CSG120/7 cells and showed that LUZP1 should function similarly in other epithelial cells (Fig 2I and J).

2. The association of LUZP1 with MT is not clear in Fig1H. Its colocalization with MTs at the AJCs should be shown by immunostaining.

Response:

Thank you for the suggestion. As mentioned above in response (2) to the main comment, we re-acquired the immunofluorescence image (Fig 1F).

3. Since Y27632 treatment reduced LUZP1 localization at CRs (Fig2G), the authors propose a positive feedback loop between ppMLC and LUZP1 localization at CRs. The authors have to consider alternative possibilities and explain what initiates MLC diphosphorylation. What recruits LUZP1 to CRs? The authors show that LUZP1 interacts with ZO1, but it is unclear what role ZO1 plays in LUZP1 recruitment to TJs or CRs.

Response:

Thank you for these valuable comments. We aimed to address your abovementioned concerns one by one as follows:

> Since Y27632 treatment reduced LUZP1 localization at CRs (Fig2G), the authors propose a positive feedback loop between ppMLC and LUZP1 localization at CRs. The authors have to consider alternative possibilities

We agree that Y27632 treatment is not sufficient to conclude that the recruitment of LUZP1 to TJ is indeed ppMLC-dependent. To address your concerns, we examined the effect of Y27632 on the global architecture of CRs and then quantified LUZP1 junctional localization before and after calyculin A treatment.

Immunostaining analyses revealed that our way of transient Y27632 treatment did not have an apparent effect on CR architecture as was revealed by MLC immunostaining (Fig EV3F). Furthermore, artificial upregulation of ppMLC levels within CRs using calyculin A was shown to significantly promote LUZP1 junctional localization (Fig 3C and D). Thus, considering that ppMLC and LUZP1 were highly co-localized within CRs (Fig 3F and G) and the binding affinity between LUZP1 and di-phosphomimetic MLC was very strong (Fig 3I–K), we concluded that the recruitment of LUZP1 to TJ should be ppMLC-dependent and that there should be a positive feedback loop between ppMLC and LUZP1 localization at CRs (Fig 3E).

> explain what initiates MLC diphosphorylation

Previous studies have showed that apical junctional complex (AJC)-localized Rho-associated coiled-coil kinase (ROCK) is the primary kinase initiating MLC phosphorylation within CRs (Sawyer et al, 2010; Suzuki et al, 2012; Martin & Goldstein, 2014; Takeichi, 2014). Our *in vitro* assays showed that LUZP1 itself could not promote MLC

phosphorylation (Fig 4C) and that LUZP1 promoted MLC phosphorylation only in the presence of protein phosphatase 1c (PP1c) (Fig 5E, F, 6D, and EV4G).

> **What recruits LUZP1 to CRs?**

We believe that TJs and ppMLC are both essential for recruiting LUZP1 to CRs because (1) the destruction of TJs resulted in the delocalization of LUZP1 from CRs (Fig 1I and EV2E) and (2) LUZP1 localized to CRs in a ppMLC-dependent manner when TJs were intact (Fig 3A and B).

> **The authors show that LUZP1 interacts with ZO1, but it is unclear what role ZO1 plays in LUZP1 recruitment to TJs or CRs.**

In the revised manuscript, we found that LUZP1 also interacted with ZO-2 in addition to ZO-1 (Fig EV2F and G). We also found that knocking out either ZO-1 or ZO-2 did not influence LUZP1 junctional localization (Figure below, A and B). Considering that the destruction of TJs induced by ZO-1/-2 DKO resulted in the delocalization of LUZP1 from TJ-associated CRs (Fig 1I and EV2E), we believe that the affinity of LUZP1 to TJs, which may be mediated by its binding affinity to both ZO-1 and ZO-2, is crucial for LUZP1 junctional localization.

4. **The authors should better explain why Merlin is a relevant substrate to assess PP1c activity. Fig5F: PP1c does not completely suppress Pak1-mediated phosphorylation of Merlin. More direct measurement of PP1c activity is warranted.**

Response:

Thank you for this important comment. Based on previous reports (Jin et al, 2006; Morrison et al, 2001), we believe that Merlin is a relevant substrate for assessing PP1c activity. We agree that it is problematic that PP1c did not completely suppress Pak1-mediated phosphorylation of Merlin in the previous manuscript. The reason behind this undesirable phenomenon appears to be that PP1c activity critically depends on the purifying and storage processes. In the revised manuscript, we performed the experiment again more carefully and subsequently confirmed that PP1c completely suppresses Pak1-mediated phosphorylation of Merlin (Fig 5F).

5. **Fig3E, Fig5EF: In vitro assay of ppMLC needs improvement, because GST-MLC bands are very weak. GST-MLC can't be 20kDa, is this a typo? The error bars are very large in Fig5EF. Additional controls, such as LUZP1 alone and, possibly Calyculin A treatment, are needed.**

Response:

Thank you for these valuable comments. As suggested, we conducted the *in vitro* direct binding assay once again, renewed the representative figure, and corrected the typo regarding the molecular weight of MLC (Fig 3I). We also performed several *in vitro* MLC phosphorylation assays again with additional controls (LUZP1 alone without PP1c β/δ ; please see the lane 5 in Fig 5E and Fig 5F). Although additional controls with calyculin A treatment were attempted,

we decided not to include them in the revised manuscript, because we failed to get consistent results (i.e., the extent that calyculin A promotes phosphorylation varied significantly).

6. Figs.6AB. Apical constriction needs to be better supporting images. Does nocodazole reduce ppMLC specifically at CRs or overall levels?

Response:

We added representative images to further support that nocodazole treatment reversed the apical constriction of REV cells in co-cultures of LUZP1 KO and Venus-LUZP1-expressing LUZP1 knockout (REV) cells (Fig EV4E). Regarding the effect of nocodazole on ppMLC levels, ppMLCs at CRs were found specifically reduced after nocodazole treatment as evidenced by immunofluorescence analyses (Fig 6B). In fact, in the stress fiber, the contrasting effect of nocodazole on ppMLC levels was observed where nocodazole upregulated ppMLC levels (Figure below, A).

Response to referee #2:

Tsukita and colleagues have previously shown a structural and functional link between tight junctions and microtubules. Here, they extend this work uncovering a new mechanistic component, LUZP1, and extend the functional significance to apical constriction, a process contributing to epithelial cell morphogenesis in columnar epithelia and neural tube closure. While LUZP1 was known to be involved in neural tube closure, there was no information of the underlying cellular and molecular mechanisms. This is an interesting and largely convincing study that should be of interest to a wide readership. The authors use an impressive set of complementary assays to support their conclusions that appear to be performed to a high standard. Occasionally, the paper requires some modifications to improve clarity and minor experiments to support the conclusions drawn.

Major Comments

- 1) All components tested to interact with LUZP1 bind to the same fragment of the protein, the C-terminal domain. This includes ZO-1, MLC as well as tubulin/microtubules. It is surprising that junctional LUZP1 should bind to all of these proteins with the same domain. However, inhibition of ROCK leads to junctional dissociation of LUZP1, despite its ZO-1 binding site, and does not lead to any obvious association with microtubules (Fig 2G). Hence, one wonders which of these interactions actually occur under physiological conditions (i.e., not transfected and vastly overexpressing HEK293 cells or in a test tube). Do these proteins all coimmunoprecipitate from wild type Eph4 cells? Is binding of MLC, ZO-1 and tubulin to LUZP1 competitive? How do the authors envision how these interactions are coordinated within the junction?

Response:

Thank you for these insightful comments. As recommended, we conduct a co-immunoprecipitation assay of wild-type (WT) Eph4 cells using the anti-LUZP1 antibody. It was found that LUZP1, MLC, α -tubulin, and ZO-1 were all co-immunoprecipitated (Fig EV11), suggesting that LUZP1 simultaneously interacts with MLC, α -tubulin, and ZO-1 under physiological conditions and that they are not competitive.

We considered that all of these interactions are essential for the physiological role of LUZP1. Specifically, we considered that both TJs and ppMLC are essential for LUZP1 to localize at TJ-associated CRs and that MTs are important for LUZP1 to fulfill its function of inhibiting protein phosphatase 1c (PP1c). The reasoning behind our consideration is as follows; first, the destruction of TJs induced by ZO-1/-2 double knockout (DKO) resulted in the delocalization of LUZP1 from TJ-associated CRs (Fig 1I and EV2E). However, it was also found that knocking out only ZO-1 did not delocalize LUZP1 from CRs (Figure below, A) and LUZP1 also interacted with ZO-2 in addition to ZO-1 (Fig EV2F and G). Therefore, we considered that the interaction between LUZP1 and TJs may be mediated by its binding affinity to both ZO-1 and ZO-2, and plays an essential role in the recruitment of LUZP1 to TJ-associated CRs. Second, when TJs were intact, transient Y27632 treatment significantly reduced LUZP1 recruitment to TJ-associated CRs (Fig 3A and B) without any apparent effect on MLC localization (Fig EV3F), and conversely, calyculin A treatment significantly promoted LUZP1 recruitment to TJ-associated CRs (Fig 3C and D), suggesting that ppMLC is the main determinant of LUZP1 localization when TJs are intact. Finally, considering that (1) MTs promoted the LUZP1-mediated inhibition of PP1c in vitro (Fig 6D and EV4G) and (2) MTs promoted ppMLC upregulation within CRs via LUZP1 in cells (Fig 6B and C), we believe that MTs plays an essential role for LUZP1 to fulfill its function.

A Cocultures of WT and ZO-1 KO cells

- 2) **Fig 1C:** According to the methods, the co-IP was performed with cold cell extract; hence, the interaction seen is likely to be with α -tubulin, not microtubules as they depolymerise in the cold. As binding sites for free tubulin and microtubules may be distinct, the co-sedimentation assay should be performed to identify the microtubule binding site within the LUZP1 fragments.

Response:

Thank you for this important suggestion. As recommended, we conducted several MT co-sedimentation assays at room temperature in the presence of taxol to identify the MT binding region within LUZP1. Surprisingly, in contrast to our previous co-immunoprecipitation results showing that the LUZP1 C-terminal was responsible for α -tubulin binding (Figure below, A), MT co-sedimentation assays revealed that the LUZP1 N-terminal and middle regions were responsible for MT binding (Figure below, B and Fig EV1E). This result was very clear and indicated that the binding sites for free tubulin and microtubules are distinct. We sincerely appreciate this comment.

In addition, we examined the possibility that the domain-binding property of LUZP1 is altered by the detergent that was included in the RIPA buffer-based co-immunoprecipitation assay but not in the MT co-sedimentation assays. MT co-sedimentation assays revealed that no LUZP1 regions could bind to MTs under RIPA buffer-based conditions. In contrast, co-immunoprecipitation assays revealed that all regions of LUZP1 were bound to α -tubulin under an HBS-based and a detergent-free condition (Figure below, C), whereas only the LUZP1 C-terminal was bound to α -tubulin under RIPA buffer-based conditions (Figure below, A). Therefore, we concluded that the detergent alters the binding property of LUZP1 regions. Considering that (1) the main aim of region analysis was determining the LUZP1 region responsible for MT binding and that (2) binding under a detergent-free condition is more similar to physiological conditions, we decided to only use the MT co-sedimentation assay-based domain analyses in the revised manuscript (Fig EV1E).

Furthermore, because of uncertainty surrounding LUZP1 region analyses under RIPA buffer-based conditions, we also reconducted co-immunoprecipitation analyses between LUZP1 domains and MLC and ZO-1 under detergent-free conditions. However, we could not sufficiently elute ZO-1 and MLC from transfected HEK293 cells without detergent and thus could not determine the binding domains of LUZP1 to MLC and ZO-1 in the absence of detergent. Because of the uncertainty of the region analysis data, we decided **not to** present the results of the co-immunoprecipitation analyses between LUZP1 regions and MLC and ZO-1 in the presence of detergent.

A Co-immunoprecipitation assay conducted in RIPA (with detergent)

B MT co-sedimentation assay conducted in an HBS-based detergent-free condition

C Co-immunoprecipitation assay conducted in an HBS-based detergent-free condition

D MT co-sedimentation assay conducted in RIPA (with detergent)

- 3) **Page 8, line 14: What is the evidence that MLC itself is still at junctions upon treatment with Y27632? This should be tested by IF.**

Response:

Thank you for the insightful comment. We acquired immunofluorescence images to confirm that MLC is still localized at the junctions after Y27632 treatment (Fig EV3F).

- 4) **Figure 3C: The authors conclude that this experiment shows binding to ppMLC; however, there is no evidence presented that the co-immunoprecipitated MLC is phosphorylated. An immunoblot for ppMLC should be shown.**

Response:

Thank you for this valuable suggestion. We conducted the co-immunoprecipitation assay to demonstrate the LUZP1 regions responsible for MLC binding and did not interpret the result as evidence for the binding between LUZP1 and ppMLC. We understand that the co-immunoprecipitated MLC is most likely not phosphorylated. Other than this co-immunoprecipitation assay, we concluded that LUZP1 directly bound to ppMLC via direct binding assays between LUZP1 and di-phosphomimetic MLC (Fig 3H–K); we also showed the high co-localization between LUZP1 and ppMLC (Fig 3F and 3G).

Furthermore, as mentioned in our response to comment 2, we reconducted this LUZP1 region analysis under an HBS-based, detergent-free condition to obtain an immunoblot for ppMLC; however, we failed to obtain sufficient results as it is difficult to elute MLC without detergent. Because of the uncertainty surrounding LUZP1 region analyses under RIPA buffer-based conditions, we decided not to present these results.

- 5) **Figure 3H-L: How does one know that slower diffusion is not an intrinsic property of ppMLC (e.g., due to being part of active myosin) as opposed to be due to binding LUZP1? The assay should also be performed in the absence of LUZP1 to justify the conclusion.**

Response:

Thank you for this insightful and valuable comment. To address this, we conducted additional fluorescent decay after photoactivation (FDAP) analysis on the bottom surface of the cells in the absence of LUZP1. We could not draw a ROI in the lateral membrane because AA-MLC and DD-MLC did not localize to the lateral membrane in the absence of LUZP1 (Figure below, A). Subsequently, we found that the diffusion property of AA-MLC and DD-MLC (supposedly from actin filaments within the stress fiber) differed and DD-MLC diffused more slowly than AA-MLC (Figure below, B). Considering that MLC localized to the lateral side only after LUZP1 was ectopically expressed (Figure below, C) and that our ROIs for FDAP analysis were marked in the lateral membrane (Figure below, D), it is still very possible that the slower diffusion of DD-MLC compared with AA-MLC results from the stronger affinity of LUZP1 to DD-MLC than to AA-MLC. However, it is difficult to completely deny the influence of the intrinsic difference in diffusion between AA- and DD-MLC from actin filaments; therefore, we decided **not** to present these results in the revised manuscript.

Minor comments

- 6) **Fig 1B:** What was the input, purified Flag-LUZP1? This should be specified to justify the conclusion that the interaction is direct.

Response:

We apologize for the oversight. Yes, it was purified FLAG-LUZP1, which we have now specified in the figure legend.

- 7) **Page 6:** The description of figure 1F-J is confusing, appears to routinely refer to the wrong panels and the panels are not in order of the text.

Response:

Thank you for bringing this to our attention. We have accordingly revised the ordering of Fig 1.

- 8) **Page 7, last paragraph:** The 'epithelial' behaviour described may be observed in some epithelial cell lines but should not be declared as being the general behaviour of epithelial cells as they come in different types and shapes, and, in vivo, may behave rather differently.

Response:

Thank you for this comment. We have revised the sentence to avoid generalizing (page 8, 2nd paragraph).

- 9) **Page 10, start of new section:** The experiments show that TJ recruitment of LUZP1 is ROCK activity-dependent. Whether it is indeed ppMLC-dependent is not shown.

Response:

Thank you for this constructive comment. We admit that experiments using Y27632 were not sufficient to conclude that the recruitment of LUZP1 to TJ is indeed ppMLC dependent. To resolve your concern, we quantified LUZP1 junctional localization before and after calyculin A treatment. We found that although the difference was minimal, artificial upregulation of ppMLC levels within CRs using calyculin A significantly promoted the junctional localization of LUZP1 (Fig 3C and D). Furthermore, a closer look at LUZP1 localization before and after nocodazole treatment indicated that LUZP1 junctional localization was apparently disturbed as junctional ppMLC levels decreased (Fig 6B

and C), although co-localization between ppMLC and LUZP1 was similar (Fig EV4F). Accordingly, we can now consider that LUZP1 recruitment to TJ-associated CRs is ppMLC-dependent but not ROCK activity-dependent.

Response to referee #3:

In this manuscript the authors describe their findings on the protein LUZP1 and its involvement in apical constriction in vertebrate epithelial cells. The authors show that LUZP1 colocalises with myosin at the level of tight junctions in epithelial cells in culture, and that cells expressing LUZP1 can apically constrict whereas K.O. cell cannot. They go on to show that LUZP1 can bind to MRLC in a phosphorylation state dependent manner, and that it inhibits myosin phosphatase through direct binding. LUZP1 was originally identified by the authors at a microtubule-associated protein, and they show that interaction with microtubules is important for LUZP1 to exert its effect on myosin levels at tight junctions.

The apical constriction of epithelial cells is a key process during morphogenesis and organ formation, and thus a better understanding of the mechanisms and variations underlying this process are key to understanding development and should be of interest to a variety of readers. Furthermore, the interplay of actomyosin and microtubules is emerging as an important, albeit understudied, theme in morphogenesis, and as such the described mechanism is also of great interest.

The experiments presented appear well-executed and controlled, and the conclusions drawn seem well supported by the data. I have a few general and specific comments, though, that I think would be worth addressing or commenting on to strengthen the paper further.

- 1) Firstly, the authors put particular emphasis on the notion that apical constriction in vertebrate cells works through a different mechanism than the one analysed in great detail in invertebrates such as *Drosophila* or *C.elegans*. In these invertebrate models apical constriction is driven by a highly dynamic pool of apical-medial actomyosin, whereas the role of junctional myosin in this process might rather be to provide a ratcheting mechanism to stabilise a shrunken apical shape. The authors state 'In contrast, in vertebrate apical constriction, actomyosin-based circumferential rings (CRs) associated with the apical junctional complex (AJC), which includes tight junctions (TJs) and adherens junctions (AJs), play a central role.' in the introduction, but without citing any studies to support this. I agree that the vast majority of support for dynamic apical-medial actomyosin driving apical constriction stems from invertebrate systems, but there are indications that similar pools of myosin with similar functionality do also operate in vertebrates (see for instance Sumigay, K. D., Terwilliger, M., & Lechler, T. (2018). Morphogenesis and Compartmentalization of the Intestinal Crypt, 1-35. <http://doi.org/10.1016/j.devcel.2018.03.024>). It took year and very careful in vivo live imaging in *Drosophila* for instance to uncover the apical-medial actomyosin networks and their functions. I would feel happier with a less dogmatic distinction between invertebrate and vertebrate apical constriction.

Response:

Thank you for the insightful comment. In the revised manuscript, we added the description that there are indications that apical-medial actomyosin do also operate in vertebrates and cited the article you mentioned (page 3, second paragraph).

- 2) I am also still somewhat confused about actomyosin-driven constriction at the level of tight junctions compared to the usually described contractile actomyosin belt at the level of adherens junctions. Are these both present in the cells analysed here, are they continuous/overlapping? Are tight junction membrane proteins really also bona-fide adhesion receptors and force transducers between cells, which this work would imply? This should be commented on, please!

Response:

Thank you for this valuable comment. As can be seen in the representative electron micrograph of Fig EV1A, actomyosin-based CRs show associations with both AJs and TJs, and TJ- and AJ-associated CRs are continuous. The point that we want to emphasize is that TJ- and AJ-associated CRs may have some distinct roles, considering that the

newly-discovered di-phosphorylated myosin light chain (ppMLC)-triggered, LUZP1-based system specifically functions at TJ-associated CRs and not at AJ-associated CRs. Therefore, although previous research primarily focused on the association between AJs and CRs for apical constriction as you mentioned, future studies should also focus on the association between TJs and CRs. We have included a sentence regarding this important point (page 13, last paragraph). Unfortunately, we did not examine whether the force was transduced between cells via TJ membrane proteins and, to the best of our knowledge, there is no evidence to date that concretely shows that TJ membrane proteins are bona-fide force transducers between cells.

- 3) **How do the authors envisage LUZP1 to function in apical constriction events during embryogenesis in vivo? Is the spatial and temporal control of apical constriction provided by the regulation of ppMLC and LUZP1 expressed in the same cells helps to amplify the myosin activation? Or is there control and regulation at the level of LUZP1 expression? In the neural tube for instance, is it only expressed when apical constriction is important for tube folding and closure?**

Response:

Thank you for this insightful comment. It is known that LUZP1 KO leads to neural tube closure defect (Hsu et al, 2008), but there are no data regarding whether spatial and temporal controls of apical constriction are mediated by the regulation of ppMLC and LUZP1 or whether there is control and regulation at LUZP1 expression level. We think this is a crucial question to answer and are willing to assess these points in the future.

- 4) **The authors identify LUZP1 as a microtubule-binding protein and then show that it also binds to MLC as well as myosin phosphatase. Do the authors have any evidence about domains that mediate these binding interactions and how likely they are to all occur at the same time?**

Response:

Thank you for the suggestion. We have conducted a series of co-immunoprecipitation assays to determine the regions that are responsible for LUZP1 binding to multiple binding partners. However, we found out that the detergent in RIPA-buffer may alter the results of region analyses and that ZO-1 and MLC could not be eluted from transfected HEK293 cells without detergent (please see our response to major comment 2 of referee #2).

To examine whether LUZP1 binding to multiple binding partners occurs at the same time in cells, we conduct co-immunoprecipitation assays of wild type Eph4 cells using an anti-LUZP1 antibody. It was found that LUZP1, MLC, α -tubulin, and ZO-1 were all co-immunoprecipitated (Fig EV1I), confirming that LUZP1 interacts with multiple binding partners at the same time in cells.

- 5) **In Figure 2B the authors quantify the difference in apical constriction of cells in culture that express tagged LUZP1 in a mutant background in comparison to the KO cells: which cells were measured in the LUZP1-KO? The graph shows a ratio for these well above 1, i.e. they are apically expanded. Is this because they are the cells next to the apically constricted ones and thus are making up for their neighbours? Otherwise I find it hard to see why and how they should achieve this apically dilated shape!**

Response:

Thank you for this valuable comment. The LUZP1 KO cells around the Venus-LUZP1-expressing LUZP1 KO (REV) cells were measured. We believe, as you mentioned, that these LUZP1 KO cells are apically expanded and making up for their apically constricted neighbors.

- 6) **The authors use a somewhat strange expression multiple times throughout the manuscript 'ppMLC mobilizes LUZP1...'. What do they mean by 'mobilizing'? I assume they mean 'recruits'?**

Response:

Thank you for bringing this to our attention. We did indeed mean "recruiting," and have accordingly changed

“mobilize” to “recruit” in the revised manuscript.

Thank you for submitting a revised version of your manuscript. Your study has now been seen by all original referees, who find that most of their main concerns have been addressed and are now broadly in favour of publication of the manuscript. There now remain only a few mainly editorial issues that have to be addressed before I can extend formal acceptance of the manuscript:

1. Please address the remaining points from reviewer #1 (I presume that in point 3 they refer to Figure 4F) and add appropriate discussion on the tubulin co-sedimentation assays as requested by reviewer #3.
2. Our publisher has done their pre-publication check on your manuscript. I have attached the file here. Please take a look at the word file and the comments regarding the figure legends and respond to the issues. Please also use this version when you resubmit the revised version.
3. Please differentiate between the contributions of Hatsuho Kanoh and Hiroka Kashihara (HKan and HKas); Tomoaki Mizuno and Takeshi Matsui (TMi and TMa) in the Author Contributions section.
4. Please include references to the Appendix Tables S1 and S2 in the manuscript text.
5. Data Availability section currently refers to "statics data". I presume that statistics are meant instead, please check.

Please let me know if you have any further questions regarding any of these points. You can use the link below to upload the revised files.

Thank you again for giving us the chance to consider your manuscript for The EMBO Journal. I am looking forward to receiving the final version.

Referee #1:

The authors have addressed many previous concerns and, in principle, the manuscript can be recommended for publication. However, there are a couple of remaining comments.

1. The authors show that LUZP1 was no longer recruited to AJC in ZO1/ZO2 dKO cells, and that ppMLC levels decreased in ZO1/ZO2 dKO cells. Is LUZP1 involved in maintaining ppMLC but not pMLC levels within TJ-associated CRs? It appears important to evaluate pMLC levels in ZO1/ZO2 dKO cells, LUZP1 OE cells or LUZP1 KO cells. In previous studies, pMLC (rather than ppMLC) was associated with increased actomyosin tension at AJ (Fanning et al., 2012; Tubota et al., 2014; Choi et al., 2016).
2. The authors confirmed that LUZP1 knockout increased ppMLC in MTD-1A and CSG1 cells. Does LUZP1 k/o also reveal apical constriction defects in these cells?
3. Fig. 4B needs anti-MLC as a control for immunoblotting.

Referee #2:

The authors have addressed all my comments in an adequate manner.

Referee #3:

The authors have carefully addressed all the reviewers' comments and have extended their experiments, especially to address the question of recruitment of LUZP1 to TJs as well as its interaction with other factors in this position. These experiments mostly help to elucidate the function further. My one main comment would be that the direct binding or association with microtubules is still more deduction than proof, as a co-IP from cold lysate bringing down α -tubulin does not prove binding to filamentous microtubules (as also pointed out by review 2), but rather to unpolymerised tubulin. And the fact that the co-sedimentation assays at RT in the presence of taxol identify different regions of LUZP1 as binding to microtubules somewhat further underlines that worry about the co-IP results. The testing of different detergent conditions also rather confuses (me) more than it helps: why should N-terminal and middle regions of LUZP1 co-sediment with filamentous MTs, but then in detergent-free co-IPs bind to monomeric α -tubulin (or maybe rather tubulin dimers)?

Overall I agree, though, that using the co-sedimentation assays for the manuscript is the right way to go. I would just urge the authors to describe their interaction findings cautiously.

The authors performed the requested editorial changes.

Point-by-point responses to the Editor and Referees:

We wish to express our strong appreciation to the Editor and Referees for their insightful comments on our paper. Our reactions to the comments are as follows.

Response to Referee #1:

Main Comment:

The authors have addressed many previous concerns and, in principle, the manuscript can be recommended for publication. However, there are a couple of remaining comments.

- 1. The authors show that LUZP1 was no longer recruited to AJC in ZO1/ZO2 dKO cells, and that ppMLC levels decreased in ZO1/ZO2 dKO cells. Is LUZP1 involved in maintaining ppMLC but not pMLC levels within TJ-associated CRs? It appears important to evaluate pMLC levels in ZO1/ZO2 dKO cells, LUZP1 OE cells or LUZP1 KO cells. In previous studies, pMLC (rather than ppMLC) was associated with increased actomyosin tension at AJ (Fanning et al., 2012; Tubota et al., 2014; Choi et al., 2016).**
- 2. The authors confirmed that LUZP1 knockout increased ppMLC in MTD-1A and CSG1 cells. Does LUZP1 k/o also reveal apical constriction defects in these cells?**
- 3. Fig. 4B needs anti-MLC as a control for immunoblotting.**

Response:

Thank you for your important comments. We tried our best to address your above-mentioned concerns one by one as follows:

- 1) The authors show that LUZP1 was no longer recruited to AJC in ZO1/ZO2 dKO cells, and that ppMLC levels decreased in ZO1/ZO2 dKO cells. Is LUZP1 involved in maintaining ppMLC but not pMLC levels within TJ-associated CRs? It appears important to evaluate pMLC levels in ZO1/ZO2 dKO cells, LUZP1 OE cells or LUZP1 KO cells. In previous studies, pMLC (rather than ppMLC) was associated with increased actomyosin tension at AJ (Fanning et al., 2012; Tubota et al., 2014; Choi et al., 2016).**

Yes, LUZP1 regulates ppMLC levels but not pMLC levels. Immunofluorescence analyses revealed that pMLC levels were similar between LUZP1 knockout (LUZP1 KO) and LUZP1-expressing [wild-type (WT) and Venus-LUZP1-expressing LUZP1 KO (REV)] cells (Figure below, A and B). Consistently, further immunofluorescence analyses revealed that, when ZO-1/-2 DKO cells fully became polarized, pMLC levels were similar between WT and ZO-1/-2 DKO cells but ppMLC levels were significantly reduced in ZO-1/-2 DKO cells (Figure below, C and D), altogether indicating that LUZP1 regulates ppMLC levels but not pMLC levels at tight junction (TJ)-associated circumferential rings (CRs).

We are aware of the fact that ZO-1/-2 knockdown (KD) or KO in MDCK cells leads to increased junction tension and upregulation of junctional pMLC levels, as you kindly mentioned with referring to three important articles (Fanning et al., 2012; Tokuda et al., 2014; Choi et al., 2016). However, it should be particularly emphasized that ZO-1/-2 KD and KO in MDCK cells also leads to the significant thickening of actomyosin-based CRs. Therefore, it is quite possible that the upregulation of junctional pMLC levels merely reflect the increased actomyosin accumulation at cell–cell junctions. Furthermore, it should also be emphasized that these articles did not assess ppMLC levels and, therefore, cannot even suggest whether pMLC or ppMLC is important for actomyosin contraction.

As we mentioned in the introduction section of our manuscript, Watanabe et al beautifully revealed that ppMLC,

not pMLC, is crucial for the contraction of actomyosin-based stress fibers whereas pMLC is important for the maintenance of stress fibers (Watanabe et al., Mol Biol Cell 2007). Similarly, Farber et al revealed that the Shroom-ROCK axis, which is a well-known driver for the contraction of CRs, upregulates junctional ppMLC levels but do not affect pMLC levels (Farber et al., Mol Biol Cell 2011). In this context, it should be natural to think that ppMLC, not pMLC, is critical for the contraction of CRs. However, we believe that rigorous researches are still required to firmly conclude the differential role of pMLC and ppMLC in the contraction of actomyosin network and this “pMLC or ppMLC” problem should confuse the reader of our manuscript; therefore, we decided not to include immunofluorescent micrographs (Figure below, A to D) in the manuscript.

2) **The authors confirmed that LUZP1 knockout increased ppMLC in MTD-1A and CSG1 cells. Does LUZP1 k/o also reveal apical constriction defects in these cells?**

Yes, LUZP1 KO MTD-1A and CSG120/7 cells display apical constriction defect. To prove this, we calculated the apical area of co-cultures of LUZP1 KO and WT cells. Subsequent statistical analyses revealed that LUZP1 KO MTD-1A cells have significantly larger apical area than WT MTD1A cells [$42.9 \pm 29.9 \mu\text{m}^2$ (WT) vs. $102.3 \pm 29.9 \mu\text{m}^2$ (LUZP1 KO), $p < 0.01$] and LUZP1 KO CSG120/7 cells have significantly larger apical area than WT CSG120/7 cells [$50.2 \pm 24.8 \mu\text{m}^2$ (WT) vs. $100.0 \pm 36.6 \mu\text{m}^2$ (LUZP1 KO), $p < 0.01$] (Fig. EV2H).

3) **Fig. 4B needs anti-MLC as a control for immunoblotting.**

As you kindly pointed out, MLC might be preferable for a control in this situation. However, since the calyculin treatment did not affect neither the level of MLC nor the level of α -tubulin (figure below, A), we believe that it can also be acceptable to use α -tubulin as a control. Due to the current COVID-19/SARS-CoV-2 pandemic, it will take more than a month to collect the high-quality data that can be used for statistical analyses; therefore, we decided to go on without changing this figure. I ask and appreciate your understanding in this case.

Response to referee #2:

Main Comment:

The authors have addressed all my comments in an adequate manner.

Response:

Thank you for your kind comments. We totally appreciate your constructive comments in the previous round of the revision because these comments substantially improved our manuscript.

Response to referee #3:

The authors have carefully addressed all the reviewers' comments and have extended their experiments, especially to address the question of recruitment of LUZP1 to TJs as well as its interaction with other factors in this position. These experiments mostly help to elucidate the function further. My one main comment would be that the direct binding or association with microtubules is still more deduction than proof, as a co-IP from cold lysate bringing down α -tubulin does not prove binding to filamentous microtubules (as also pointed out by review 2), but rather to unpolymerised tubulin. And the fact that the co-sedimentation assays at RT in the presence of taxol identify different regions of LUZP1 as binding to microtubules somewhat further underlines that worry about the co-IP results. The testing of different detergent conditions also rather confuses (me) more than it helps: why should N-terminal and middle regions of LUZP1 co-sediment with filamentous MTs, but then in detergent-free co-IPs bind to monomeric α -tubulin (or maybe rather tubulin dimers)?

Overall I agree, though, that using the co-sedimentation assays for the manuscript is the right way to go. I would just urge the authors to describe their interaction findings cautiously.

Response:

Thank you for your constructive comments. We agree that the co-immunoprecipitation and co-sedimentation results (with or without the detergent) which we presented in the previous round of the revision were complicated but we believe that the reason behind those complicated results was that 1) the binding sites for free tubulin and microtubules are distinct and 2) the detergent probably alters the binding property of LUZP1 regions. We also agree that we have to state the result of co-immunoprecipitation assays of WT Eph4 cells cautiously (Fig. EV11); therefore, we changed the term “confirmed” to “indicated” in the sentence mentioning about this experiment (page 6, middle of 2nd paragraph).

Editor accepted the manuscript.

Corresponding Author Name: Sachiko Tsukita

Journal Submitted to: Embo journal

Manuscript Number: EMBOJ-2020-104712R